# Comparative chloroplast genomes and phylogenetic relationships of *Aglaonema modestum* and five variegated cultivars of *Aglaonema*

**Dong-Mei Li**●*, **Gen-Fa Zhu***, **Bo Yu, Dan Huang**

Guangdong Key Lab of Ornamental Plant Germplasm Innovation and Utilization, Environmental Horticulture Research Institute, Guangdong Academy of Agricultural Sciences, Guangzhou, 510640, China

* biology.li2008@163.com (D-ML); genfazhu@163.com (G-FZ)

**Data Availability Statement:** Data Availability Statement: The six newly assembled chloroplast genomes in this study have been deposited in GenBank with accession numbers MK262737,

## Abstract

*Aglaonema*, commonly called Chinese evergreens, are widely used for ornamental purposes. However, attempts to identify *Aglaonema* species and cultivars based on leaf morphology have been challenging. In the present study, chloroplast sequences were used to elucidate the phylogenetic relationships of cultivated *Aglaonema* in South China. The chloroplast genomes of one green species and five variegated cultivars of *Aglaonema*, *Aglaonema modestum*, 'Red Valentine', 'Lady Valentine', 'Hong Yan', 'Hong Jian', and 'Red Vein', were sequenced for comparative and phylogenetic analyses. The six chloroplast genomes of *Aglaonema* had typical quadripartite structures, comprising a large single copy (LSC) region (91,092–91,769 bp), a small single copy (SSC) region (20,816–26,501 bp), and a pair of inverted repeat (IR) regions (21,703–26,732 bp). The genomes contained 112 different genes, including 79–80 protein coding genes, 28–29 tRNAs and 4 rRNAs. The molecular structure, gene order, content, codon usage, long repeats, and simple sequence repeats (SSRs) were generally conserved among the six sequenced genomes, but the IR-SSC boundary regions were significantly different, and 'Red Vein' had a distinct long repeat number and type frequency. For comparative and phylogenetic analyses, *Aglaonema costatum* was included; it was obtained from the GenBank database. Single-nucleotide polymorphisms (SNPs) and insertions/deletions (indels) were determined among the seven *Aglaonema* genomes studied. Nine divergent hotspots were identified: *trnH-GUG-CDS1_psbA*, *trnS-GCU_trnS-CGA-CDS1*, *rps4-trnT-UGU*, *trnF-GAA-ndhJ*, *petD-CDS2-rpoA*, *ycf1-ndhF*, *rps15-ycf1-D2*, *ccsA-ndhD*, and *trnY-GUA-trnE-UUC*. Additionally, positive selection was found for *rpl2*, *rps2*, *rps3*, *ycf1* and *ycf2* based on the analyses of Ka/Ks ratios among 16 Araceae chloroplast genomes. The phylogenetic tree based on whole chloroplast genomes strongly supported monophyletic *Aglaonema* and clear relationships among Aroideae, Lasioideae, Lemnoideae, Monsteroideae, Orontioideae, Pothoideae and Zamioculcadoideae in the family Araceae. By contrast, protein coding gene phylogenies were poorly to strongly supported and incongruent with the whole chloroplast genome phylogenetic tree. This study provided valuable genome resources and helped identify *Aglaonema* species and cultivars.

MK262738, OK094434, OK094435, OK094436, and OK094427.

**Funding:** This research was financially supported by the Guangdong Basic and Applied Basic Foundation Project (No. 2021A1515010893), Special Financial Fund of Foshan-High-level Guangdong Agricultural Science and Technology Demonstration City Project (2022), and Guangdong Province Modern Agriculture Industry Technical System-Flower Innovation Team Construction Project (2021KJ121). The funders had no role in study design, data collection and analysis, decision to publish, or preparation of the manuscript.

**Competing interests:** The authors have declared that no competing interests exist.

## Introduction

*Aglaonema*, commonly called Chinese evergreens, belongs to the family Araceae and comprises 21 species [1–3]. They are native to southeast Asia, northeast India and southern China southward through Malaysia, New Guinea and the Philippines [1]. *Aglaonema* is a reliable ornamental crop for commercial growers, and plants readily adapt to low light and low relative humidity levels encountered under interior conditions [1,3–5]. *Aglaonema* cultivation in the United States started in the 1930s and soon became highly successful in Florida [2]. In China, *Aglaonema modestum* has been widely cultivated for ornamental and medical purposes in southern provinces. Historically, most new *Aglaonema* cultivars were introduced directly from the wild or were from established cultivars [3]. Over the past 20 years, both public and private breeders worldwide have generated many new *Aglaonema* cultivars by hybridization and tissue-cultured mutation selection [2–5].

Although *Aglaonema* has become a crucial ornamental foliage genus, identifying *Aglaonema* species and cultivars based on leaf morphology has been challenging; for example, *Aglaonema* 'Red Valentine' is closely analogous to *Aglaonema* 'Lady Valentine' in leaf morphology. Both are very popular in South China because of their beautiful foliage, good growth habit and excellent performance indoors. Commercially, tissue culture propagation is used to speed 'Red Valentine' plants. In our previous study, *Aglaonema* 'Hong Yan' and *Aglaonema* 'Hong Jian' are two mutations found among a population of tissue-cultured 'Red Valentine' plants. Although the genetic relationships of *Aglaonema* species and cultivars have been investigated using amplified fragment length polymorphism (AFLP) markers [2], the results have been limited in high resolution for *Aglaonema* species and cultivar identification. Recently, phylogenetic relationships among 13 Aroideae species have been evaluated using whole chloroplast genomes [6]. However, only one complete chloroplast genome of *Aglaonema costatum* has been reported in the genus *Aglaonema* [6], hindering molecular identification of *Aglaonema* species and cultivars, particularly failing to distinguish popular *Aglaonema* species and variegated cultivars in South China, namely, *A. modestum*, 'Red Valentine', 'Lady Valentine' and 'Red Vein'. Therefore, we attempted to report six complete chloroplast genomes of these *Aglaonema* species and five cultivars, identify them, and explore their phylogenetic relationships.

Chloroplast genomes of angiosperms are highly conserved and have a typical quadripartite structure containing a large single copy (LSC) region, a small single copy (SSC) region, and two copies of inverted repeats (IRs) [7,8]. The size of chloroplast genomes ranges from 107 kb (*Cathaya argyrophylla*) to 280 kb (*Pelargonium*) and generally comprises 110–130 genes encoding ribosomal RNAs (rRNAs), transfer RNAs (tRNAs) and proteins [6–9]. The rapid development of high-throughput sequencing methods has made chloroplast genome sequencing faster and of higher quality. Complete chloroplast genomes have been widely used for phylogenetic analyses and molecular marker development in higher plants [6,9–11].

In this study, we sequenced and analyzed the chloroplast genomes of one species and five cultivars of *Aglaonema*: *A. modestum*, 'Red Valentine', 'Hong Yan', 'Hong Jian', 'Red Vein', and 'Lady Valentine'. Our study aimed first to analyze the whole chloroplast genome structure characteristics among the tested *Aglaonema* species and cultivars. Second, we examined variations in long repeats and microsatellites among the six newly sequenced *Aglaonema* chloroplast genomes. Third, we screened divergent hotspot regions for use as potential DNA markers in genus *Aglaonema*. Fourth, we inferred the phylogenetic relationship of one species and five cultivars of *Aglaonema* along with one published species from the GenBank database (*A. costatum*, MN046881) and phylogenetic position in the family Araceae using the complete chloroplast genome and protein coding gene sequence alignments, respectively.

## Materials and methods

### Plant materials and DNA isolation

The fresh and healthy leaves of one green species and five variegated cultivars, namely, *A. modestum*, 'Red Valentine', 'Hong Yan', 'Hong Jian', 'Lady Valentine', and 'Red Vein' (Fig 1), were collected from the resource garden of the Environmental Horticulture Research Institute (23°23′N, 113°26′E) at the Guangdong Academy of Agricultural Sciences, Guangzhou, Guangdong Province, China. *A. modestum* is erect with a dark green stem and leaves, which is not variegated (Fig 1A). Both 'Red Valentine' and 'Lady Valentine' are variegated, with green and red along the midrib, along veins and throughout the leaf blade (Fig 1B and 1C). The leaves of 'Hong Jian' are narrow, oblong, dark pink to dark red, accounting for approximately 75% to 90% of the leaf area, and the shape of the apex of the leaves is strongly acute, with green and dark yellow green blotches at the margin and throughout the leaf blade (Fig 1D). The leaves of 'Hong Yan' are ovate-cordate, dark red, accounting for approximately 95% of the leaf area, and the shape of the apex of the leaves is obtuse to moderately acute, with green blotches at the leaf margin (Fig 1E). The leaves of 'Red Vein' are dark green with pink to red along the midrib and veins (Fig 1F). The collected leaf samples were quickly frozen in liquid nitrogen and then stored at -80°C until use. Genomic DNA was extracted using a modified sucrose gradient centrifugation method [12]. The DNA integrity of the extracted genomic DNA was examined by 1% agarose gel electrophoresis, and the concentration was checked using a NanoDrop 2000 microspectrometer (Wilmington, DE, USA).

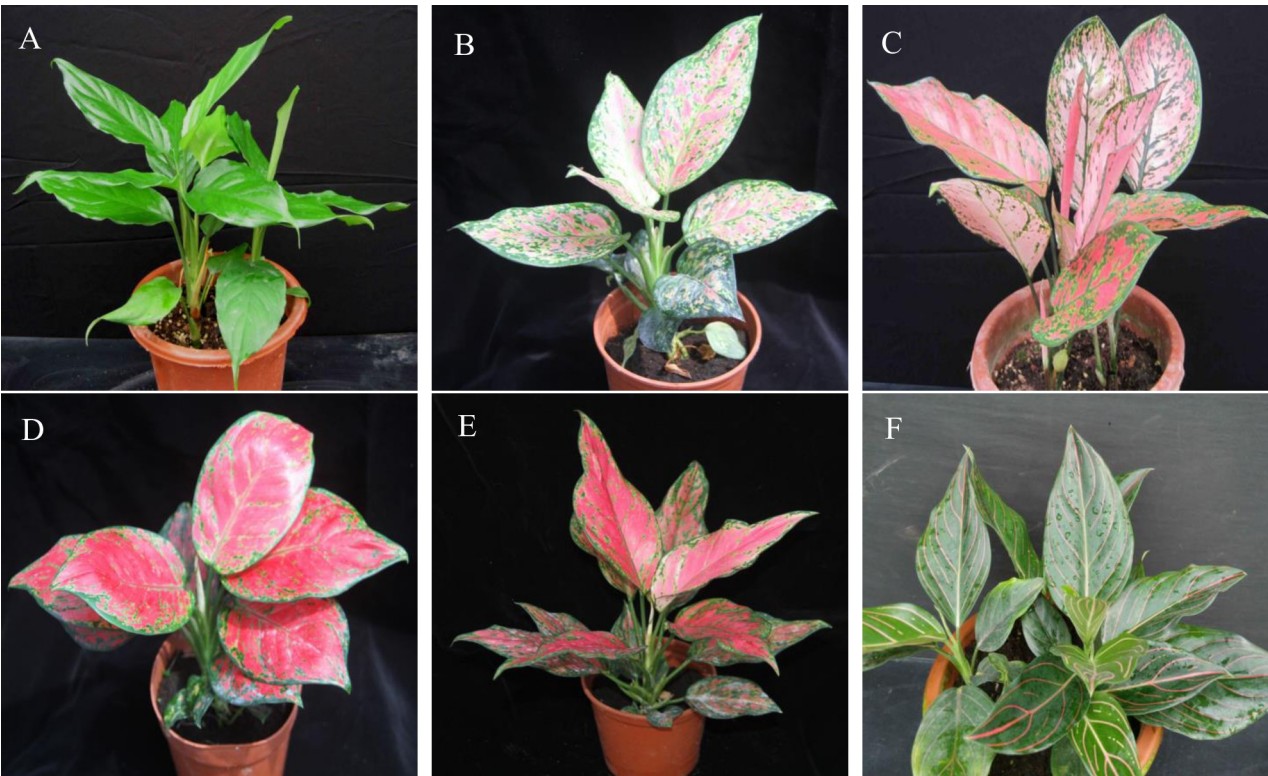

**Fig 1. Comparison of leaf morphologies among green species and variegated cultivars of genus *Aglaonema*.** (A) *Aglaonema modestum*, (B) *Aglaonema* 'Red Valentine', (C) *Aglaonema* 'Lady Valentine', (D) *Aglaonema* 'Hong Yan', (E) *Aglaonema* 'Hong Jian', and (F) *Aglaonema* 'Red Vein'.

## Chloroplast genome sequencing, assembly, annotation and structure analysis

For two variegated cultivars, 'Lady Valentine' and 'Red Vein', two libraries with insert sizes of 300 bp and 10 kb were constructed after purification and then sequenced on an Illumina HiSeq X Ten instrument (Biozeron, Shanghai, China) and a PacBio Sequel platform (Biozeron, Shanghai, China), respectively. Regarding the other four samples (*A. modestum*, 'Red Valentine', 'Hong Yan', and 'Hong Jian'), a library with insert sizes of 300 bp was constructed for each sample after purification and then sequenced on an Illumina HiSeq X Ten instrument (Biozeron, Shanghai, China).

After sequencing, all raw reads were filtered using Trimmomatic v.0.3 with default parameters to obtain clean reads [13]. The assembly of the chloroplast genomes of 'Lady Valentine' and 'Red Vein' (Illumina data and PacBio data) was performed using SOAPdenovo v.2.04 software [14]. The clean Illumina sequences of the other four samples were assembled into complete chloroplast genomes using SOAPdenovo v.2.04 with default parameters [14] and the chloroplast genome of *A. costatum* (MN046881) as the reference. The GC content was calculated using Geneious version 11.0.4 software [15]. The genes of the chloroplast genomes were annotated using the online server Dual Organellar Genome Annotator (DOGMA) with default parameters [16]. BLAST searches for the complete chloroplast genome were performed against the following public databases: nonredundant protein databases, SwissProt, Gene Ontology, Clusters of Orthologous Groups, and Kyoto Encyclopedia of Genes and Genomes. Finally, verification of tRNAs and rRNAs was performed using tRNAscanSE with default parameters [17]. The map of the chloroplast genome was drawn using Organellar Genome DRAW v1.3.1 with default parameters [18]. The protein coding sequences of each sequenced chloroplast genome were extracted, and the relative synonymous codon usage (RSCU) values were calculated using MEGA7 software [19]. The clustered heatmap of RSCU values of six sequenced *Aglaonema* chloroplast genomes was drawn using R v.3.6.3 [20].

## Analyses of long repeats and microsatellites

REPuter software (available online: http://bibiserv.techfak.uni-bielefeld.de/reputer/) [21] was used to identify and analyze the size and positions of long repeats, including the forward, palindrome, reverse and complement repeat units, within the six newly sequenced *Aglaonema* chloroplast genomes. The long repeat detection settings were a minimum repeat size of 30 bp and a Hamming distance of 3. MIcroSAtellite (MISA) was used to identify SSRs in the six newly sequenced *Aglaonema* chloroplast genomes (available online: http://pgrc.ipk-gatersleben.de/misa/) [22]. SSR detection parameters were set as follows: unit-size (nucleotide) _min-repeats: 1_8, 2_5, 3_4, 4_3, 5_3, 6_3.

## Comparative genome and sequence divergence analyses

To calculate nucleotide variability (Pi) among seven chloroplast genomes within *Aglaonema*, six newly sequenced and one obtained from GenBank (*A. costatum*, MN046881), protein coding, intron and intergenic regions among these seven chloroplast genomes were extracted and then calculated using DnaSP version 6 software [23]. The sliding window length was set to 600 bp, and the step size was set to 200 bp. mVISTA software (http://genome.lbl.gov/vista/mvista/about.shtml) in the Shuffle-LAGAN mode [24] was used to compare the above seven complete chloroplast genomes of *Aglaonema*. The nonsynonymous (Ka) and synonymous (Ks) substitution rates of chloroplast genomes in seven members of *Aglaonema* and nine other members of Araceae (S1 Table) were estimated using KaKs_Calculator [25] with default parameters. The

Pi values, mVISTA and Ka/Ks were analyzed using the complete chloroplast genome of *A. modestum* as the reference sequence. Based on genome annotations, the borders among the IR, LSC and SSC regions of the seven *Aglaonema* chloroplast genomes were also compared.

First, the seven *Aglaonema* complete genomes were aligned using MUMmer software [26] and adjusted manually where necessary using Se-Al 2.0 (available online: http://tree.bio.ed.ac.uk/software) [27], using the annotated *A. modestum* chloroplast genome as the reference. The single nucleotide polymorphisms (SNPs) and insertion/deletions (indels) were recorded separately, as well as their locations in the chloroplast genome. Second, the five variegated cultivars were analyzed to identify SNPs and indels using the annotated 'Red Valentine' chloroplast genome as the reference. Third, 'Lady Valentine' and 'Red Vein' were compared and analyzed to identify SNPs and indels.

### Phylogenetic analyses

A total of 57 complete chloroplast genome sequences were used for phylogenetic analyses, including 6 newly sequenced genomes, 50 previously reported Araceae genomes, and *Acorus americanus* (EU273602), which was set as the outgroup taxa (S1 Table). The previously reported genome sequences were downloaded from GenBank database. The protein coding sequences of these 57 genomes were extracted using Geneious version 11.0.4 software [15]. Both the complete genome sequences and protein coding sequences were used for maximum likelihood (ML) tree construction. The nucleotide sequences were aligned using the MAFFT plugin [28] in Geneious version 11.0.4 with default settings. The complete alignment was used to construct an ML tree using PhyML version 3.0 [29]. The general-time-reversible, gramma distribution, and invariable sites (GTR+G+I) DNA substitution model was selected, and all the branch nodes were calculated with 1000 bootstrap replicates. Bootstrap values were classified as strong ($> 85\%$), moderate (70–85%), weak (50–70%), or poor ($< 50\%$) [30].

### Results

#### General features of the six chloroplast genomes of *Aglaonema*

The total length of the six newly sequenced chloroplast genomes of *Aglaonema* ranged from 164,261 bp ('red vein') to 165,824 bp ('Hong Yan'). The chloroplast genome size of the green species *A. modestum* was 165,626 bp, which was 198 bp shorter than that of 'Hong Yan' and 1365 bp longer than that of 'Red Vein'. Interestingly, both 'Red Valentine' and 'Hong Jian' had the same genome size, which was 27 bp smaller than that of 'Hong Yan' (Tables 1 and S2). All six *Aglaonema* chloroplast genomes showed a typical quadripartite structure (Fig 2, Table 1 and S1 Fig), comprising a pair of inverted repeat (IR) regions (21,703–26,732 bp) separated by an LSC region (91,092–91,769 bp) and an SSC region (20,816–26,501 bp).

All six *Aglaonema* chloroplast genomes had GC contents varying from 35.73% to 35.91% (Tables 1 and S2). The IR region accounted for the highest GC content (41.62–41.98%), followed by the LSC region (33.90–34.00%), while the SSC region had the lowest GC content (28.49–30.17%) (Tables 1 and S2). The GC content of the protein coding regions varied slightly from 37.69% to 37.90%. The GC content at the third codon position (29.59–29.87%) was lower than that at the first (45.28–45.48%) and second (38.13–38.36%) positions in the protein coding genes of these six *Aglaonema* chloroplast genomes (S2 Table). The complete annotated chloroplast genomes were submitted to the GenBank database (accession numbers OK094437 for *A. modestum*, OK094434 for 'Red Valentine', OK094435 for 'Hong Jian', OK094436 for 'Hong Yan', MK262737 for 'Lady Valentine', and MK262738 for 'Red Vein') (Table 1).

The chloroplast genome annotation revealed 112 different genes in the six chloroplast genomes of *Aglaonema* (Table 1). Four different rRNA genes were identified in every

**Table 1. Characteristics of six newly sequenced chloroplast genomes of genus *Aglaonema*.**

| Genome characteristics | *modestum* | 'Red Valentine' | 'Hong Yan' | 'Hong Jian' | 'Lady Valentine' | 'Red Vein' |
|---|---|---|---|---|---|---|
| Genome size (bp) | 165,626 | 165,797 | 165,824 | 165,797 | 164,417 | 164,261 |
| LSC length (bp) | 91,269 | 91,092 | 91,092 | 91,092 | 91,135 | 91,769 |
| SSC length (bp) | 20,893 | 26,501 | 26,501 | 26,501 | 21,706 | 20,816 |
| IR length (bp) | 26,732 | 21,703 | 21,730 | 21,703 | 25,788 | 25,838 |
| Total genes (different) | 132 (112) | 132 (112) | 132 (112) | 132 (112) | 132 (112) | 132 (112) |
| Protein coding genes (different) | 87 (79) | 87 (79) | 87(79) | 87 (79) | 87 (80) | 87 (80) |
| tRNA genes (different) | 37 (29) | 37 (29) | 37 (29) | 37 (29) | 37 (28) | 37 (28) |
| rRNA genes (different) | 8 (4) | 8 (4) | 8 (4) | 8 (4) | 8 (4) | 8 (4) |
| Genes with introns | 18 | 18 | 18 | 18 | 19 | 19 |
| Genome GC (%) | 35.83 | 35.74 | 35.73 | 35.74 | 35.87 | 35.91 |
| Protein coding regions GC (%) | 37.69 | 37.69 | 37.71 | 37.69 | 37.72 | 37.90 |
| LSC GC (%) | 33.98 | 34.00 | 34.00 | 34.00 | 33.98 | 33.90 |
| SSC GC (%) | 29.12 | 28.53 | 28.49 | 28.53 | 29.25 | 30.17 |
| IR GC (%) | 41.62 | 41.67 | 41.67 | 41.67 | 41.98 | 41.78 |
| GenBank accession no. | OK094437 | OK094434 | OK094436 | OK094435 | MK262737 | MK262738 |

*Aglaonema* chloroplast genome. However, the different protein coding genes and different tRNA genes in each *Aglaonema* chloroplast genome were unevenly distributed. The four *Aglaonema* chloroplast genomes (*A. modestum*, 'Red Valentine', 'Hong Yan', and 'Hong Jian') all comprised 79 different protein coding genes and 29 different tRNA genes, while the other two *Aglaonema* chloroplast genomes ('Lady Valentine' and 'Red Vein') both contained 80 different protein coding genes and 28 different tRNA genes (Tables 1, 2 and S3). Although most of the protein coding genes and tRNAs in the six newly sequenced *Aglaonema* chloroplast genomes were similar, slight differences were observed. For example, regarding protein coding genes, both chloroplast genomes of 'Lady Valentine' and 'Red Vein' had the *ycf68* gene, while the other four sequenced genomes in this study lost this gene (S3 Table and S1 Fig). Furthermore, for tRNAs, the chloroplast genomes of 'Lady Valentine' and 'Red Vein' had one copy each of *trnG-UCC* and *trnI-CAU*, respectively, while *trnS-CGA*, *trnS-GGA* and *trnT-GGU* were missing; *trnS-CGA*, *trnS-GGA* and *trnT-GGU* showed one copy in the remaining four sequenced *Aglaonema* chloroplast genomes (Tables 2 and S3).

Regarding the genes duplicated in both IR regions in the chloroplast genomes of 'Lady Valentine' and 'Red Vein', 18 genes were identified, respectively, including seven protein coding genes (*ndhB*, *rpl2*, *rpl23*, *rps7*, *rps12*, *ycf2*, and *ycf68*), seven tRNA genes (*trnI-CAU*, *trnL-CAA*, *trnV-GAC*, *trnI-GAU*, *trnA-UGC*, *trnR-ACG*, and *trnN-GUU*), and all four rRNAs (*rrn4.5*, *rrn5*, *rrn16* and *rrn23*) (S1 Fig and S3 Table). Regarding the genes in the IR regions in the remaining four chloroplast genomes, the total numbers of genes were the same as those in the above two genomes. However, *ycf1* and *trnM-CAU* were present in the IR regions in the remaining four genomes, and *ycf68* and *trnI-CAU* were absent in IR regions in the remaining four genomes.

In both chloroplast genomes of 'Lady Valentine' and 'Red Vein', 17 genes (*trnA-UGC*, *trnI-GAU*, *trnG-UCC*, *trnK-UUU*, *trnL-UAA*, *trnV-UAC*, *atpF*, *ndhA*, *ndhB*, *rpoC1*, *petB*, *petD*, *rpl2*, *rpl16*, *rps12*, *rps16* and *ycf68*) contained one intron, while *ycf3* and *clpP* each contained two introns (S4 Table). Among the 19 intron-containing genes, 5 genes (*trnA-UAC*, *trnI-GAU*, *rpl2*, *ndhB* and *ycf68*) occurred in both IRs, 12 genes (*trnG-UCC*, *trnK-UUU*, *trnL-UAA*, *trnV-UAC*, *atpF*, *rpoC1*, *petB*, *petD*, *rpl16*, *rps16*, *ycf3* and *clpP*) were distributed in the LSC,

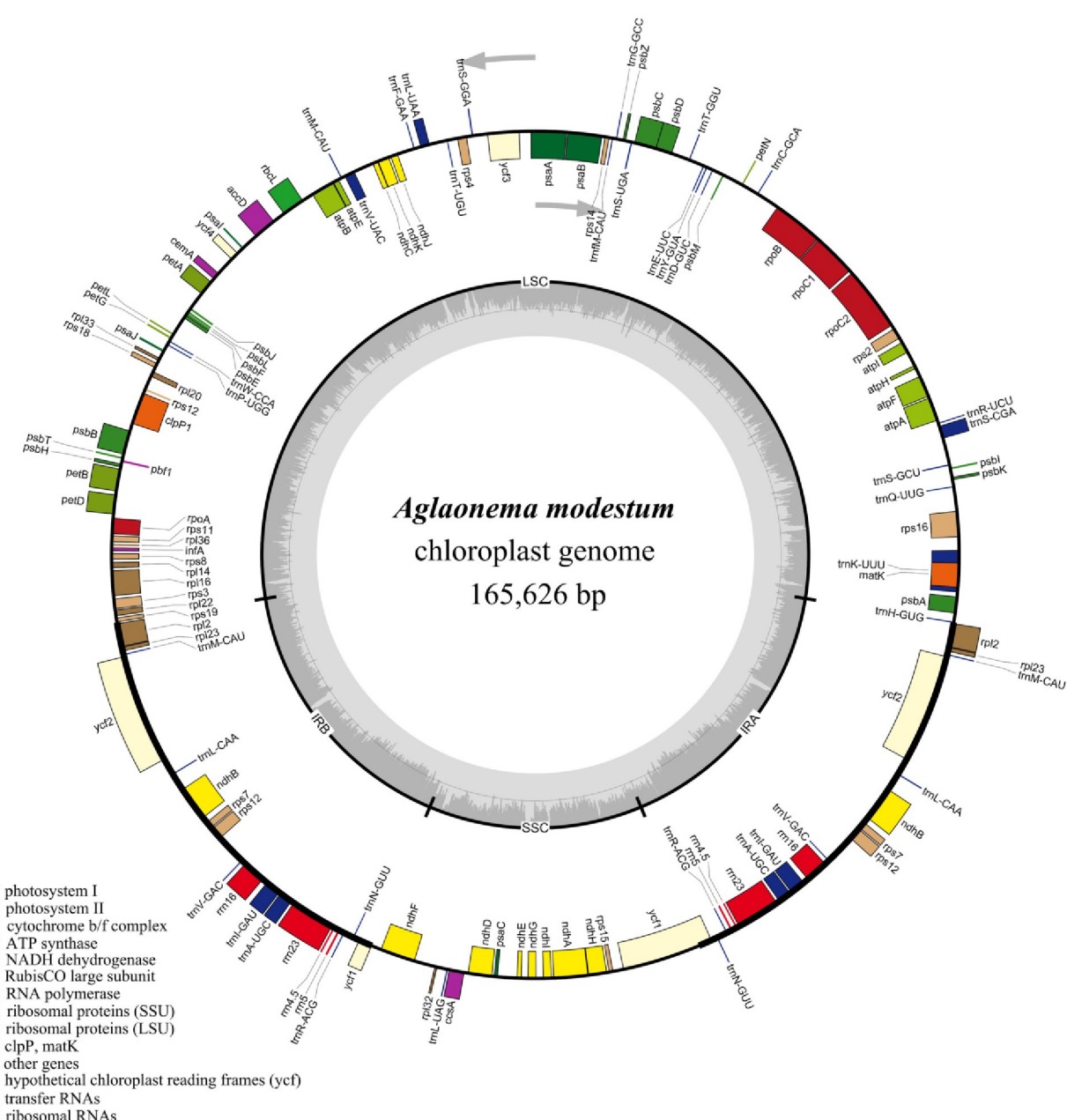

**Fig 2. Chloroplast genome map of *Aglaonema modestum*.** The gray arrowheads indicate the direction of the genes. Genes shown inside the circle are transcribed clockwise and those outside are transcribed counterclockwise. Different genes are color coded. The innermost darker gray corresponds to GC content, whereas the lighter gray corresponds to AT content. IR, inverted repeat; LSC, large single copy region; SSC, small single copy region.

one gene (*ndhA*) was in the SSC region, and one gene (*rps12*) was located in its first exon in the LSC region and the other two exons were located in both IR regions (S1 Fig and S4 Table). In comparison, in the remaining four sequenced genomes, *trnS-CGA* had one intron occurring in LSC regions, while no *ycf68* genes and no introns were observed in *trnG-UCC* (S4 Table). Therefore, only 18 genes contained introns in the remaining four genomes, respectively.

**Table 2. Genes present in the six newly sequenced chloroplast genomes of genus *Aglaonema*.**

| Category | Function | Genes |
|---|---|---|
| **Photosynthesis** | Photosystem I | *psaA, psaB, psaC, psaI, psaJ* |
| | Photosystem II | *psbA, psbB, psbC, psbD, psbE, psbF, psbH, psbI, psbJ, psbK, psbL, psbM, psbN, psbT, psbZ* |
| | Cytochrome b/f | *petA, petB\*, petD\*, petG, petL, petN* |
| | ATP synthase | *atpA, atpB, atpE, atpF\*, atpH, atpI* |
| | NADH dehydrogenase | *ndhA\*, ndhB* (×2)*\*, ndhC, ndhD, ndhE, ndhF, ndhG, ndhH, ndhI, ndhJ, ndhK* |
| | Rubisco | *rbcL* |
| **Self-replication** | RNA polymerase | *rpoA, rpoB, rpoC1\*, rpoC2* |
| | Large subunit ribosomal proteins | *rpl2* (×2)*\*, rpl14, rpl16\*, rpl20, rpl22, rpl23* (×2)*, rpl32, rpl33, rpl36* |
| | Small subunit ribosomal proteins | *rps2, rps3, rps4, rps7* (×2)*, rps8, rps11, rps12* (×2)*\*, rps14, rps15, rps16\*, rps18, rps19* (×2) |
| | Ribosomal RNAs | *rrn4.5* (×2)*, rrn5* (×2)*, rrn16* (×2)*, rrn23* (×2) |
| | Transfer RNAs | *trnA-UGC* (×2)*\*, trnC-GCA, trnD-GUC, trnE-UUC, trnF-GAA, trnfM-CAU, trnG-GCC, trnG-UCC\*&, trnH-GUG, trnI-GAU* (×2)*\*, trnI-CAU&, trnK-UUU\*, trnL-CAA* (×2)*, trnL-UAA\*, trnL-UAG, trnM-CAU* (×3)*, trnN-GUU* (×2)*, trnP-UGG, trnQ-UUG, trnR-ACG* (×2)*, trnR-UCU, trnS-GCU, trnS-CGA\*&&, trnS-GGA&&, trnS-UGA, trnT-GGU&&, trnT-UGU, trnV-GAC* (×2)*, trnV-UAC\*, trnW-CCA, trnY-GUA* |
| **Others** | Other proteins | *accD, ccsA, cemA, clpP\*\*, infA, matK* |
| | Proteins of unknown function | *ycf1* (×2)*, ycf2* (×2)*, ycf3\*\*, ycf4, ycf68* (×2)*\*&* |

×2: Gene with two copies; ×3: Gene with three copies

*: Each gene containing only one intron

**: Each gene containing two introns

&: Only present in chloroplast genomes of 'Lady Valentine' and 'Red Vein'

&&: Not present in chloroplast genomes of 'Lady Valentine' and 'Red Vein'.

## Statistics of codon usage

The six newly sequenced chloroplast genomes of *Aglaonema* were analyzed to investigate the codon usage, amino acid frequency, and relative synonymous codon usage (RSCU) (S5 Table). The total number of codons ranged from 25,800 ('Lady Valentine') to 26,670 (*A. modestum*) in six sequenced chloroplast genomes of *Aglaonema*. Although the total number of codons showed a tiny change, the types of codons and amino acids were the same. Methionine (Met) and tryptophan (Trp) were encoded by one codon usage, showing no codon bias both with RSCU values of 1.00, while others were encoded by multiple synonymous codons, ranging from two to six (Fig 3A). The codons with the four lowest RSCU values (AGC, GGC, CTG, and CGC) and three with the highest RSCU values (AGA, GCT, and TTA) were identified in the six newly sequenced genomes of *Aglaonema* (Fig 3B and S5 Table). Except for Met and Trp, thirty-two codons showed codon usage bias with RSCU > 1.00 in the chloroplast genes of all six newly sequenced genomes of *Aglaonema* (S5 Table). Interestingly, of the 32 codons, thirty-one were A/T-ending codons. Therefore, our RSCU results demonstrated that all six newly sequenced genomes had a higher usage frequency for A/T-ending than G/C-ending. Similar phenomena were found in *Epipremnum aureum* [31], *Monstera adansonii* [9], and *Rhaphidophora amplissima* [9].

## Long repeats and SSR analyses

REPuter software was used to detect four different types of long repeats, including forward, complement, reverse, and palindromic repeats. The six sequenced chloroplast genomes had

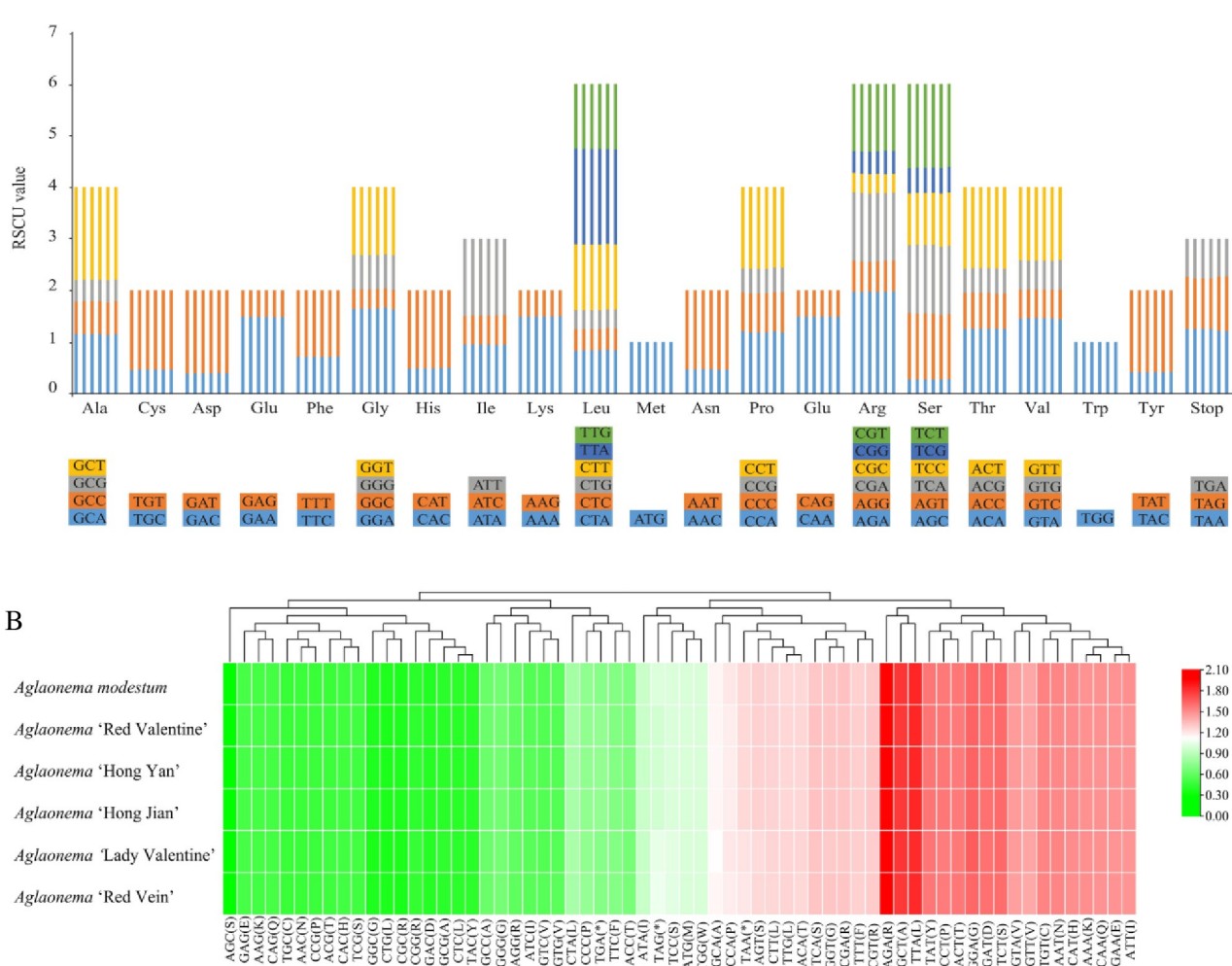

**Fig 3. Codon content of all protein coding genes of six newly sequenced *Aglaonema* chloroplast genomes.** (A) Codon content and codon usage of 20 amino acids and stop codons of all protein coding genes. Each histogram from left to right was *A. modestum*, 'Red Valentine', 'Hong Yan', 'Hong Jian', 'Lady Valentine', and 'Red Vein', respectively. (B) Heat map analysis for codon distribution of all protein coding genes in six newly sequenced chloroplast genomes. * indicates stop codons.

579 long repeats, which contained 150 forward repeats, 76 complement repeats, 158 reverse repeats, and 195 palindromic repeats (Fig 4A and S6 Table). Among the six sequenced chloroplast genomes, 'Red Vein' had the largest number (126), and 'Lady Valentine' had the smallest number (55), of long repeats (Fig 4A and S6 Table). The number of forward repeats varied from 14 ('Lady Valentine') to 37 ('Red Vein'), the number of complement repeats varied from 4 ('Lady Valentine') to 22 ('Red Vein'), the number of reverse repeats varied from 15 ('Lady Valentine') to 34 ('Red Vein'), and the number of palindromic repeats varied from 22 ('Lady Valentine') to 38 (*A. modestum*) (Fig 4A and S6 Table). The length of long repeats varied among the six sequenced chloroplast genomes, but most of the long repeats existed in the range of 30–34 bp (Fig 4B). The LSC region comprised more long repeats than the SSC and IR regions, and the details of the long repeats are also provided (S6 Table).

The SSRs in the six newly sequenced chloroplast genomes of *Aglaonema* were analyzed using MISA (Fig 5 and S7 Table). The number of detected SSRs ranged from 239 ('Lady Valentine') to 255 (*A. modestum*) (Fig 5A and S7 Table). Six types of SSRs were identified, including mononucleotide, dinucleotide, trinucleotide, tetranucleotide, pentanucleotide and

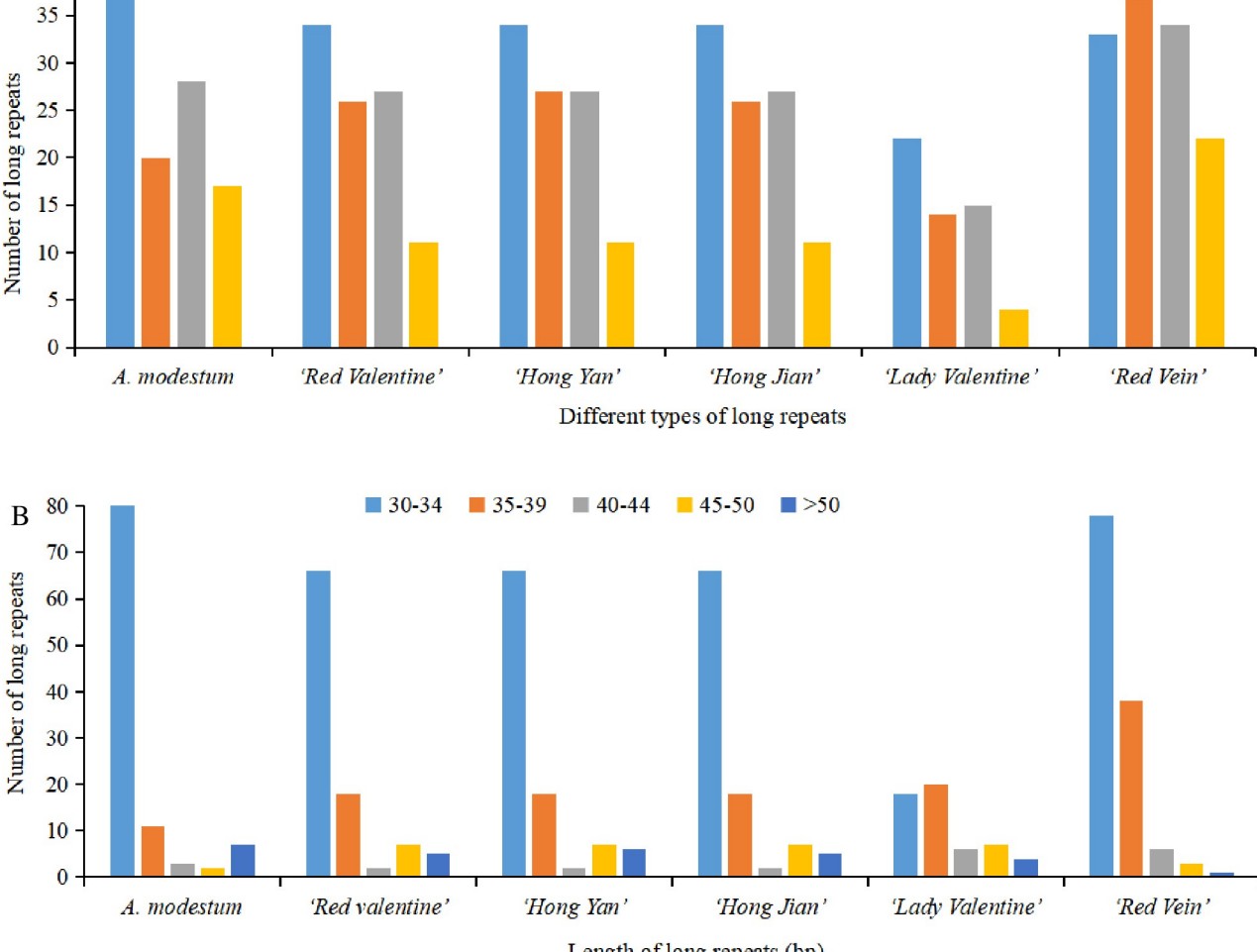

**Fig 4. Analysis of long repeat sequences in six newly sequenced *Aglaonema* chloroplast genomes.** (A) Total number of four long repeat types. (B) Length distribution of long repeats in each sequenced chloroplast genome.

hexanucleotide. Among these SSRs, only the chloroplast genomes of *A. modestum* and 'Red Vein' had hexanucleotide repeats (Fig 5A and S7 Table). Among the six sequenced chloroplast genomes, mononucleotide repeats were the most frequent, with numbers ranging from 174 to 182, followed by dinucleotides ranging from 25 to 33, tetranucleotides ranging from 15 to 20, trinucleotides ranging from 8 to 13, pentanucleotides ranging from 6 to 16, and hexanucleotides ranging from 0 to 7 (Fig 5A and S7 Table). Most of the mononucleotide SSRs were A/T repeats, which accounted for 68.63–70.71% of all SSRs among the six newly sequenced chloroplast genomes (Fig 5B and S7 Table). Regarding dinucleotide repeats, AT/AT repeats were observed most frequently, accounting for 9.68–13.11% of all SSRs (Fig 5B and S7 Table). Concerning the tetranucleotide category, the AAAT/ATTT repeats were the most abundant type, accounting for 2.42–3.92% of all SSRs (Fig 5B and S7 Table). Most SSRs were located in the LSC regions (162–176 loci; 66.39–70.97%) rather than in the SSC regions (37–47 loci; 14.92–19.26%) and IR regions (17–19 loci; 6.85–7.79%) of the six sequenced chloroplast genomes (Fig 5C and S7 Table). Additionally, SSRs were analyzed in the protein coding regions (exon, protein coding exon), intron regions and intergenic regions of the six sequenced chloroplast

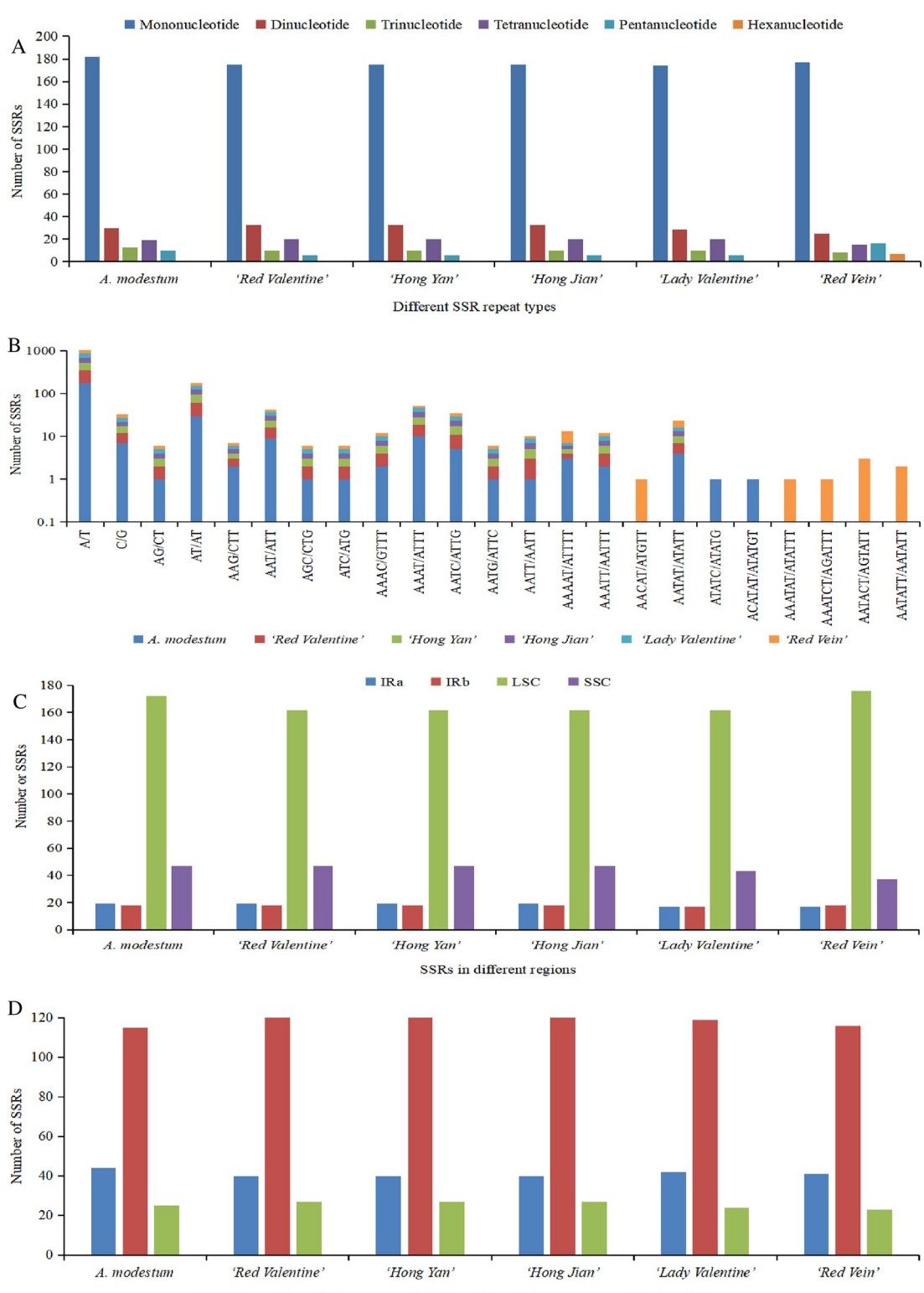

**Fig 5. Distribution of SSRs in six newly sequenced *Aglaonema* chloroplast genomes.** (A) Number of different SSR types detected in the six chloroplast genomes. (B) Frequency of identified SSR motifs in different repeat class types. (C) Frequency of SSRs in the LSC, IR and SSC regions. (D) Frequency of SSRs in the protein coding, intergenic and intron regions.

genomes, indicating that these six sequenced chloroplast genomes comprised 115 to 120 SSRs in intergenic regions, 40 to 44 SSRs in protein coding regions, and 23 to 27 SSRs in introns (Fig 5D and S7 Table). The gene *ycf1* contained more SSR loci than the other genes in all six genomes (S7 Table).

## IR contraction and expansion analyses

Comparison details of the IR-LSC and IR-SSC boundaries were performed among the six sequenced chloroplast genomes of *Aglaonema* and one published chloroplast genome of *A. costatum* (Fig 6). Regarding IRa/LSC borders, the *rpl2* and *trnH-GUG* genes were located at the boundaries of the IRa/LSC borders in all 7 *Aglaonema* chloroplast genomes. The distances between the ends of *rpl2* and IRa/LSC borders ranged from 52 bp to 62 bp (Fig 6). *trnH-GUG* expanded into the IRa regions in all 7 *Aglaonema* chloroplast genomes, with distances ranging from 15 bp to 22 bp from the IRa/LSC borders (Fig 6). Among the 7 *Aglaonema* chloroplast genomes, the *rps19* and *rpl2* genes were located in the boundaries of the LSC/IRb regions, respectively (Fig 6). A total of 21–22 bp were found between the ends of *rps19* and the LSC/IRb borders among the 7 *Aglaonema* chloroplast genomes, and the distances between the starts of *rpl2* and the LSC/IRb boundaries ranged from 53 bp to 63 bp (Fig 6).

The SSC/IRa border was located in the *ycf1* coding region, which crossed into the IRa region in 5 *Aglaonema* chloroplast genomes, including *A. modestum*, 'Red Valentine', 'Hong Yan', 'Hong Jian', and *A. costatum*. However, the lengths of *ycf1* in the IRa regions varied among the 5 *Aglaonema* chloroplast genomes, ranging from 399 bp to 868 bp (Fig 6). Two chloroplast genomes of 'Lady Valentine' and 'Red Vein', *ycf1* and *trnN-GUU*, were found in the SSC/IRa borders; the distances between the ends of *ycf1* and the SSC/IRa borders were 163

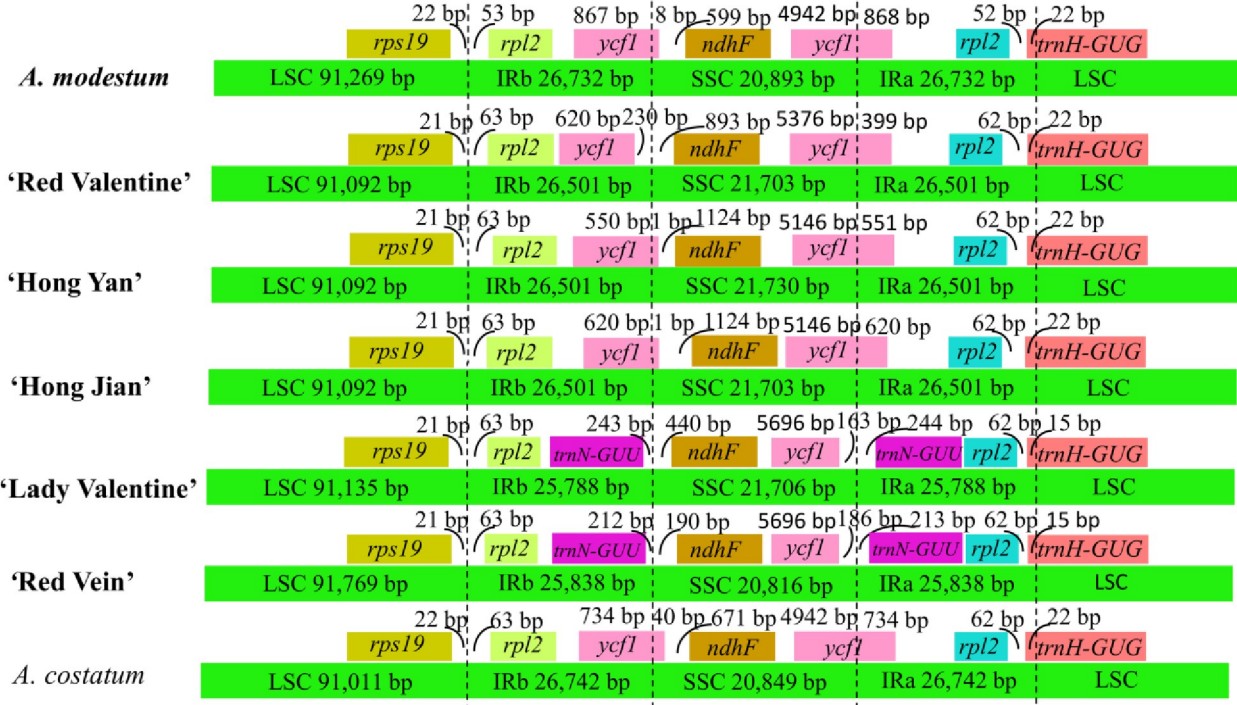

**Fig 6. Comparison of the borders of the LSC, SSC, and IR regions among seven *Aglaonema* chloroplast genomes.** The six newly sequenced *Aglaonema* chloroplast genomes in this study are in bold.

bp and 186 bp, respectively; and the distances between the starts of *trnN-GUU* and the SSC/ IRa borders were 244 bp and 213 bp, respectively (Fig 6). The IRb/SSC borders of 4 chloroplast genomes (*A. modestum*, 'Hong Yan', 'Hong Jian', and *A. costatum*) were all located in *ycf1*, and *ycf1* expanded into the SSC regions by 8 bp, 1 bp, 1 bp, and 40 bp, respectively (Fig 6). A total of 599 bp, 1124 bp, 1124 bp, and 671 bp were found between the starts of *ndhF* and the ends of *ycf1* in *A. modestum*, 'Hong Yan', 'Hong Jian', and *A. costatum*, respectively (Fig 6). Regarding 'Red Valentine', a 230 bp distance was identified between the end of *ycf1* and the IRb/SSC border and an 893 bp distance between the start of *ndhF* and the IRb/SSC border (Fig 6). Regarding 'Lady Valentine' and 'Red Vein', *trnN-GUU* and *ndhF* were located at the borders of IRb/SSC in both chloroplast genomes. A total of 243 bp and 212 bp distances were found between the ends of the *trnN-GUU* and IRb/SSC borders in 'Lady Valentine' and 'Red Vein', respectively. Furthermore, 440 bp and 190 bp were found between the starts of *ndhF* and the IRb/SSC borders in these two chloroplast genomes, respectively (Fig 6). *trnN-GUU* was duplicated in IR/SSC borders in the two genomes, 'Lady Valentine' and 'Red Vein', respectively, and *ycf1* was duplicated in IR/SSC borders in the other five *Aglaonema* genomes. Overall, the IR/LSC boundary regions of the 7 chloroplast genomes of *Aglaonema* were highly conserved, but the IR/SSC boundary regions exhibited variations.

## High variation regions and divergence analyses

The highly divergent regions in the 7 chloroplast genomes of *Aglaonema* were analyzed using DnaSP by sliding window analysis (Fig 7 and S8 Table). The IR regions displayed lower variability than the LSC and SSC regions. Twelve regions that showed obviously higher Pi values (> 0.021) and were located in the LSC and SSC regions, which included *trnH-GUG-CDS1*, *trnH-GUG-CDS1_psbA*, *trnS-GCU_trnS-CGA-CDS1*, *psbC-trnS-UGA*, *rps4-trnT-UGU*, *trnF-GAA-ndhJ*, *psbF-psbE*, *petD-CDS2-rpoA*, *ycf1-ndhF*, *rps15-ycf1-D2*, *ccsA-ndhD*, and *trnY-GUA-trnE-UUC* (Fig 7 and S8 Table).

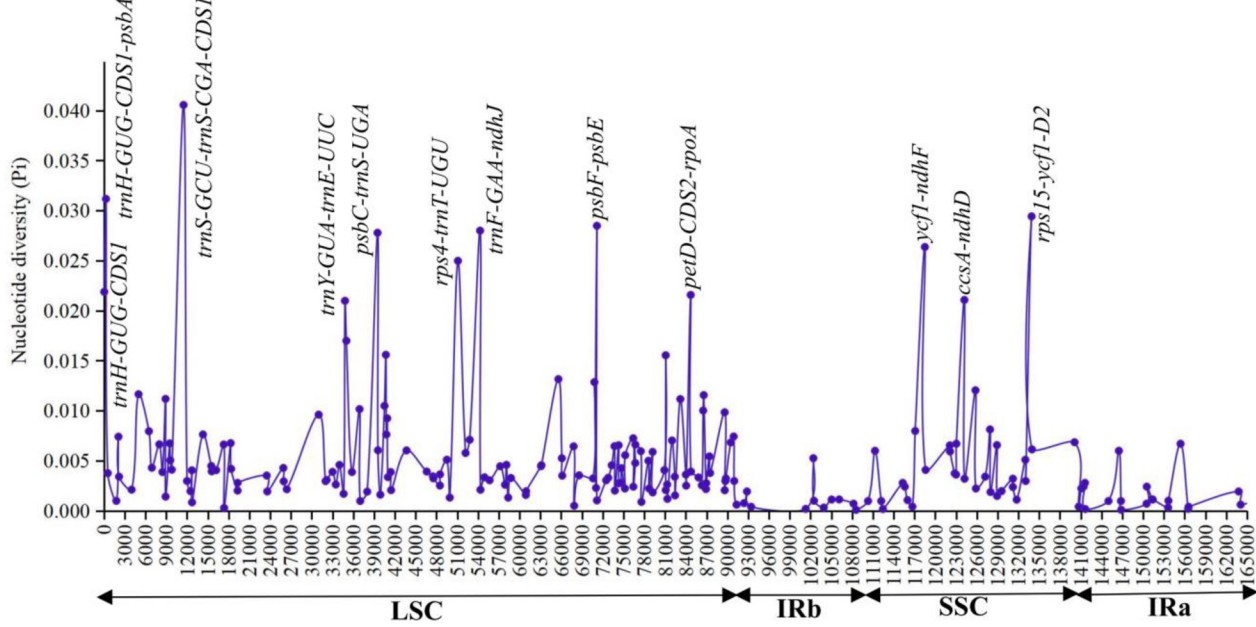

**Fig 7. Plot of sliding window analysis of the whole chloroplast genomes for nucleotide diversity (Pi) compared among seven chloroplast genomes of *Aglaonema*.** Peak regions with a Pi value of > 0.021 were labeled with loci tags of names. x-axis shows the position of the midpoint of each window. y-axis shows Pi values of nucleotide diversity in a a sliding window analysis with window length 800 bp and step size 200 bp.

Furthermore, the 7 whole chloroplast genome alignments were examined using mVISTA (Fig 8). Most of the sequence variations were found in the LSC and SSC regions, while the two IR regions showed high sequence conservation of the 7 *Aglaonema* genomes. The main divergent regions were *trnH-GUG-CDS1*, *trnH-GUG-CDS1_psbA*, *trnS-GCU_trnS-CGA-CDS1*, *rps4-trnT-UGU*, *trnF-GAA-ndhJ*, *petD-CDS2-rpoA*, *ycf1-ndhF*, and *rps15-ycf1-D2* (Fig 8). Combining the Pi values and mVISTA results, as well as considering the length of regions ≥ 200 bp for the selection of potential molecular markers of *Aglaonema*, 9 regions were found: *trnH-GUG-CDS1_psbA*, *trnS-GCU_trnS-CGA-CDS1*, *rps4-trnT-UGU*, *trnF-GAA-ndhJ*, *petD-CDS2-rpoA*, *ycf1-ndhF*, *rps15-ycf1-D2*, *ccsA-ndhD*, and *trnY-GUA-trnE-UUC* (S8 Table).

## Ka/Ks ratios among Araceae species

Using the 79 protein-coding gene sequences of *A. modestum* as references, Ka/Ks ratios were computed in the *A. modestum* chloroplast genome, six other chloroplast genomes of

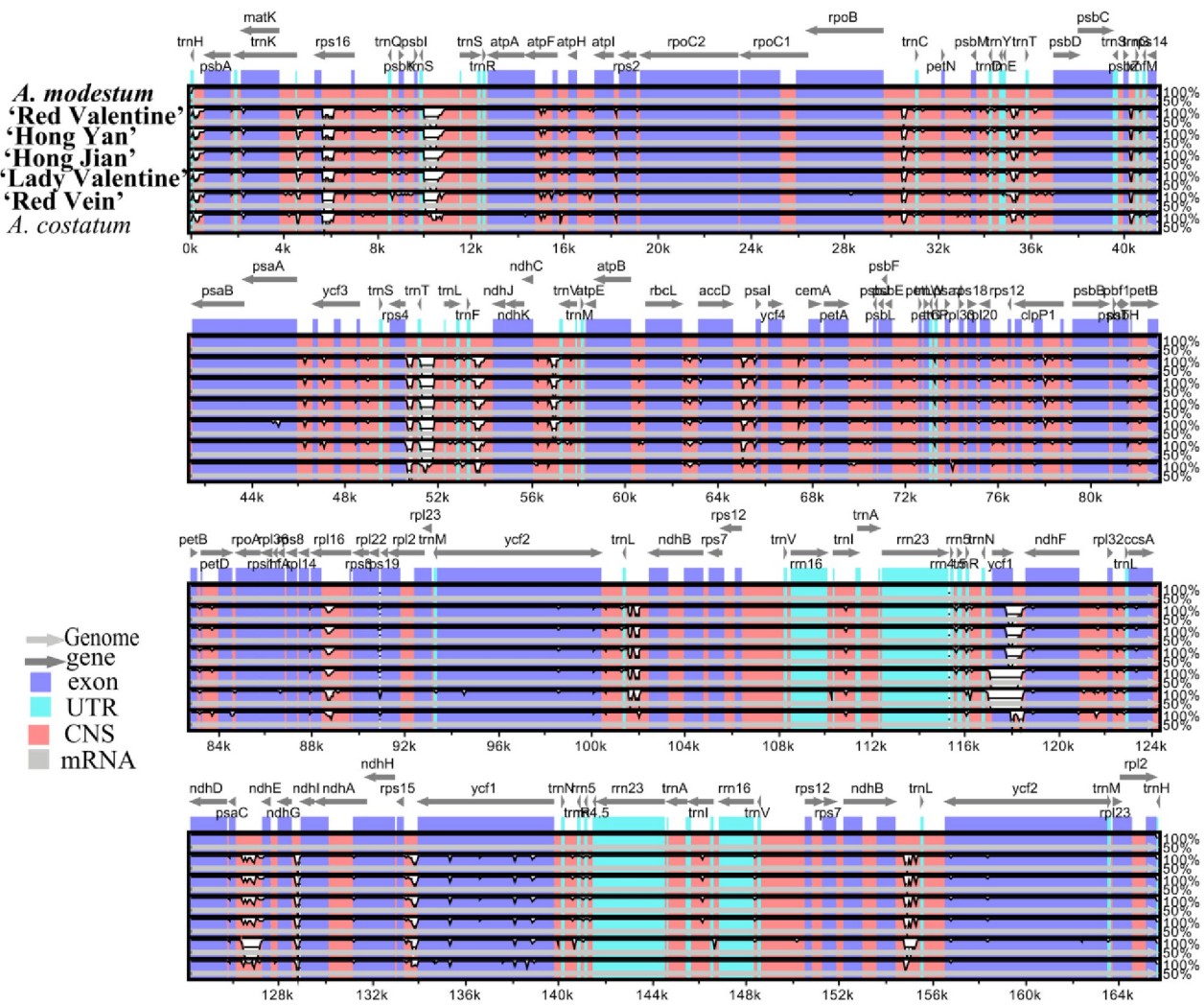

**Fig 8. Complete chloroplast genome comparison of seven *Aglaonema* chloroplast genomes using *A. modestum* as a reference.** Gray arrows and thick black lines above the alignment indicate gene orientation. Purple bars represent exons, sky-blue bars represent untranslated regions (UTRs), red bars represent non-coding sequences (CNS), gray bars represent mRNA and white regions represent sequence differences among analyzed chloroplast genomes. The y-axis represents the identity percentage ranging from 50% to 100%. The six sequenced *Aglaonema* chloroplast genomes here are in bold.

*Aglaonema* and nine other chloroplast genomes of Araceae (S9 Table). The Ka/Ks ratios were less than 1.00 and invalid for most comparison pairs (98.5%) (S9 Table). *rpl2*, *rps2*, *rps3*, *ycf1* and *ycf2* with Ka/Ks >1.00 and p < 0.05 were detected, indicating that these genes underwent positive selection (S9 Table). Furthermore, the Ka/Ks ratios of *ycf2* in five pairwise comparisons, *ycf1* in four pairwise comparisons, and *rps2* in four pairwise comparisons were all > 1.00, indicating that the three genes *ycf2*, *ycf1*, and *rps2* showed adaptation evolution under different environments.

## SNPs and indels analyses of *Aglaonema* chloroplast genomes

First, using the *A. modestum* chloroplast genome as the reference, SNP/indel loci of the chloroplast genomes of 'Lady Valentine', 'Red Vein', *A. costatum*, 'Red Valentine', 'Hong Yan', and 'Hong Jian' were detected. Three comparisons, *A. modestum* vs. 'Red Valentine', *A. modestum* vs. 'Hong Yan', and *A. modestum* vs. 'Hong Jian', revealed the same numbers of SNPs and indels, with 311 protein coding gene SNPs, 531 intergenic SNPs, and 153 indels, respectively (Fig 9, Tables 3, S10 and S11). Additionally, between *A. modestum* and 'Lady Valentine', 322 protein coding gene SNPs, 531 intergenic SNPs, and 154 indels were detected (Tables 3 and S11, and Fig 9), and the lengths of indels were mainly between 1 and 5 bp (Fig 9). Regarding *A. modestum* vs. 'Red Vein', 335 protein coding gene SNPs, 535 intergenic SNPs, and 142 indels were identified (Fig 9, Tables 3, S10 and S11). Concerning *A. modestum* vs. *A. costatum*, 333 protein coding gene SNPs, 452 intergenic SNPs, and 152 indels were found (Fig 9, Tables 3, S10 and S11).

Second, to detect SNPs/indels among five variegated cultivars of *Aglaonema*, using the 'Red Valentine' chloroplast genome as the reference, SNP/indel loci of the chloroplast genomes of 'Lady Valentine', 'Hong Yan', 'Hong Jian', and 'Red Vein' were analyzed. Two comparisons, 'Red Valentine' vs. 'Hong Yan' and 'Red Valentine' vs. 'Hong Jian', had no SNPs/indels. Fourteen SNPs and 1 indel were detected between 'Red Valentine' and 'Lady Valentine' (Fig 9, Tables 3 S10, and S11). Interestingly, these 14 SNPs were all in *psaA* (S10 Table), suggesting that *psaA* can be used to identify these two cultivars. One hundred sixty-four and 422 SNPs were detected between 'Red Valentine' and 'Red Vein' in protein coding genes and intergenic regions, respectively (Tables 3 and S10). Eighty indels were detected between 'Red Valentine' and 'Red Vein' (Fig 9, Tables 3 and S11). Third, SNPs/indels between 'Lady Valentine' and 'Red Vein' were also tested. A total of 173 and 426 SNPs in protein coding genes and intergenic regions were identified between 'Lady Valentine' and 'Red Vein', respectively (Tables 3 and S10). Eighty-two indels were also found between these two chloroplast genomes (Fig 9, Tables 3 and S11). The SNPs and indels analyzed in this study can be used for *Aglaonema* species and cultivar identification and phylogeny in the future.

## Phylogenetic analyses

Two phylogenetic trees were constructed by ML method using whole chloroplast genomes and protein coding genes, respectively (Fig 10A and 10B). *A. americanus* was classified in the family Acoraceae [32], which was used as the outgroup. The 7 subfamilies within Araceae, including Aroideae, Lasioideae, Lemnoideae, Monsteroideae, Orontioideae, Pothoideae, and Zamioculcadoideae, were strongly identified using whole chloroplast genomes with bootstrap values of 89–100% (Fig 10A). By contrast, the phylogenetic tree constructed using protein coding genes showed moderate (bootstrap value = 84%) to strong identifications (bootstrap value = 100%) (Fig 10B). Within subfamily Aroideae, *Aglaonema* was strongly supported as monophyletic (bootstrap values = 100%) (Fig 10A and 10B). *Aglaonema* was seen as either sister to *Anchomanes* (bootstrap value = 100%), and *Aglaonema* + *Anchomanes* and *Zantedeschia*

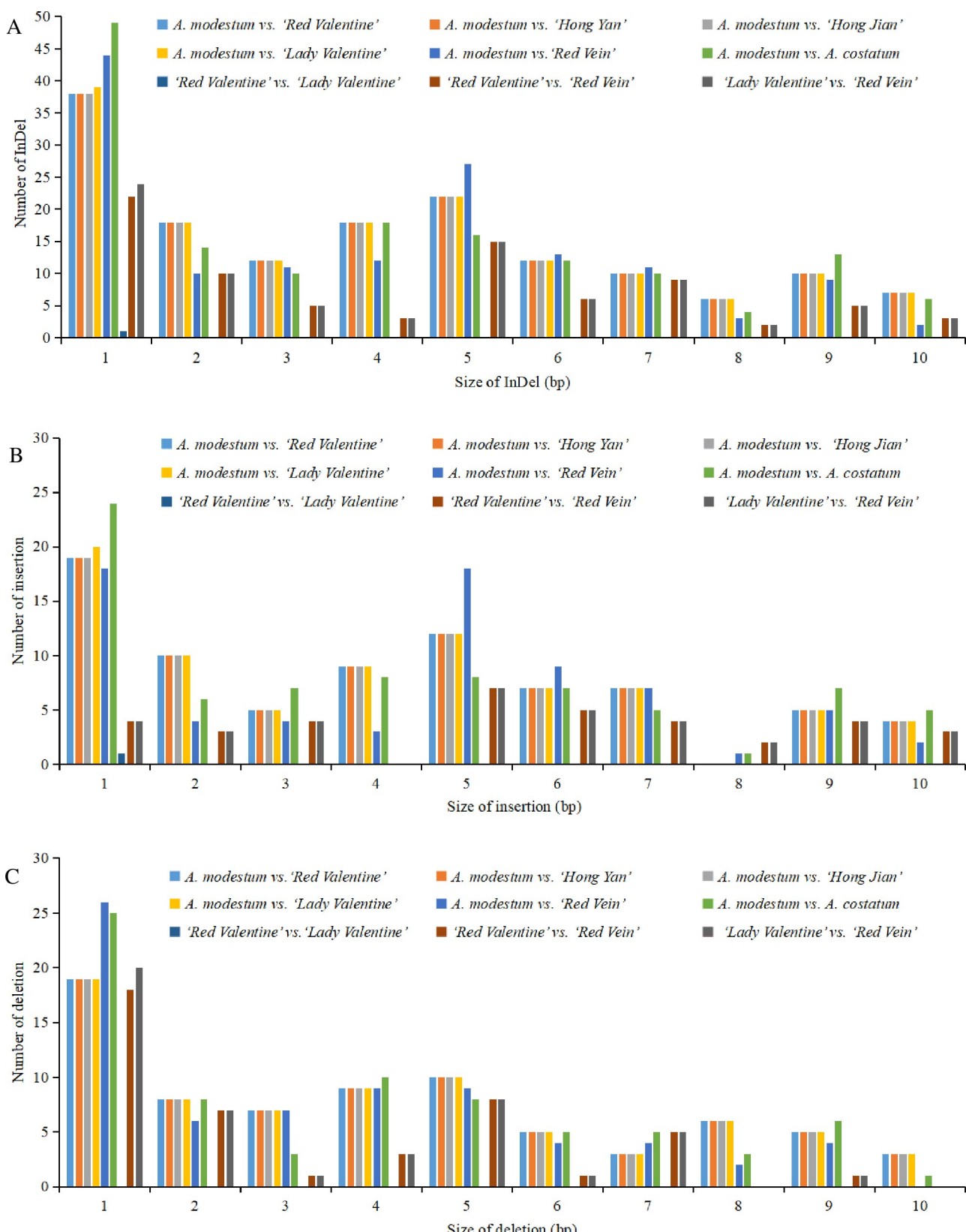

**Fig 9. Indels statistics of seven *Aglaonema* chloroplast genomes.** First, the *A. modestum* chloroplast genome was used as the reference sequence for indels analyses for the other six *Aglaonema* chloroplast genomes. Second, the five variegated cultivars were compared using chloroplast genome of 'Red valentine' as the reference. Third, 'Lady valentine' and 'Red vein' were compared. (**A**) Total indels statistics. (**B**) Insertion statistics. (**C**) Deletion statistics.

were strongly supported as sister genera (bootstrap value = 88%) (Fig 10A) or formed a poorly supported clade with *Stylochaeton* (bootstrap value = 17%) that was sister to the genera *Anchomanes* + *Zantedeschia* (bootstrap value = 100%) (Fig 10B). Within the genus *Aglaonema*, 'Hong Jian', 'Red Valentine', 'Hong Yan', and 'Lady Valentine' were clustered together with moderate to strong support (bootstrap values = 80–100%), and 'Red Vein' was sister to these four variegated cultivars with strong support (bootstrap value = 100%) (Fig 10A). *A. costatum* was sister to these five variegated cultivars with strong support (bootstrap value = 99%). *A. modestum* was sister to the clade of "*A. costatum* + five variegated cultivars" with strong support (bootstrap value = 100%) based on complete chloroplast genomes (Fig 10A). These seven *Aglaonema* species and cultivars showed similar topologies based on protein coding genes, but 'Hong Jian' and 'Red Valentine' were poorly supported in the ML analysis (bootstrap value = 0) (Fig 10B). The other five *Aglaonema* species and cultivars were strongly supported in ML analysis based on protein coding genes (bootstrap values = 89–100%) (Fig 10B).

## Discussion

In the current study, the six newly assembled chloroplast genomes of *Aglaonema* species and variegated cultivars were highly conserved, possessing typical quadripartite structures and similar GC contents (35.73–35.91%) (Fig 2, Table 1 and S1 Fig). These quadripartite structures and similar GC contents were consistent with other reported Araceae plant chloroplast genomes, such as *A. costatum* [6], *M. adansonii* [9], *E. aureum* [31], *Anthurium huixtlense* [33], and *Symplocarpus renifolius* [34,35]. Interestingly, the variation in the chloroplast genome size of three variegated cultivars, 'Hong Jian', 'Red Valentine', and 'Hong Yan', was not significant. 'Hong Jian' and 'Hong Yan' were two mutations derived from tissue-cultured 'Red Valentine' plants (Fig 1). Both 'Hong Jian' and 'Red Valentine' showed the same genome size, while 'Hong Yan' was only 27 bp longer than these two cultivars (Table 1 and S1 Fig). Similar phenomena were also found in *Hyacinthus* and *Utricularia* cultivars [36,37]. In *Hyacinthus* cultivars, 'Woodstock', 'Delft Blue' and 'Aiolos' displayed the same chloroplast genome size of 154,640 bp [36]. In the case of two *Utricularia amethystina* cultivars, Purple and Yellow, only a 95 bp difference was found [37]. These chloroplast genome sizes were not changed substantially, possibly because the chloroplast genomes did not undergo recombination among the tested cultivars [36].

**Table 3. Distribution of SNPs and indels among seven *Aglaonema* chloroplast genomes.**

| Comparison pairs | Protein coding genes SNPs | Intergenic regions SNPs | Total SNPs | Insertions | Deletions | Indels |
|---|---|---|---|---|---|---|
| *A. modestum* *vs*. 'Red Valentine' | 311 | 531 | 842 | 78 | 75 | 153 |
| *A. modestum* *vs*. 'Hong Yan' | 311 | 531 | 842 | 78 | 75 | 153 |
| *A. modestum* *vs*. 'Hong Jian' | 311 | 531 | 842 | 78 | 75 | 153 |
| *A. modestum* *vs*. 'Lady Valentine' | 322 | 531 | 853 | 79 | 75 | 154 |
| *A. modestum* *vs*. 'Red Vein' | 335 | 535 | 870 | 71 | 71 | 142 |
| *A. modestum* *vs*. *A. costatum* | 333 | 452 | 785 | 78 | 74 | 152 |
| 'Red Valentine' *vs*. 'Lady Valentine' | 14 | 0 | 14 | 1 | 0 | 1 |
| 'Red Valentine' *vs*. 'Red Vein' | 164 | 422 | 586 | 36 | 44 | 80 |
| 'Lady Valentine' *vs*. 'Red Vein' | 173 | 426 | 599 | 36 | 46 | 82 |

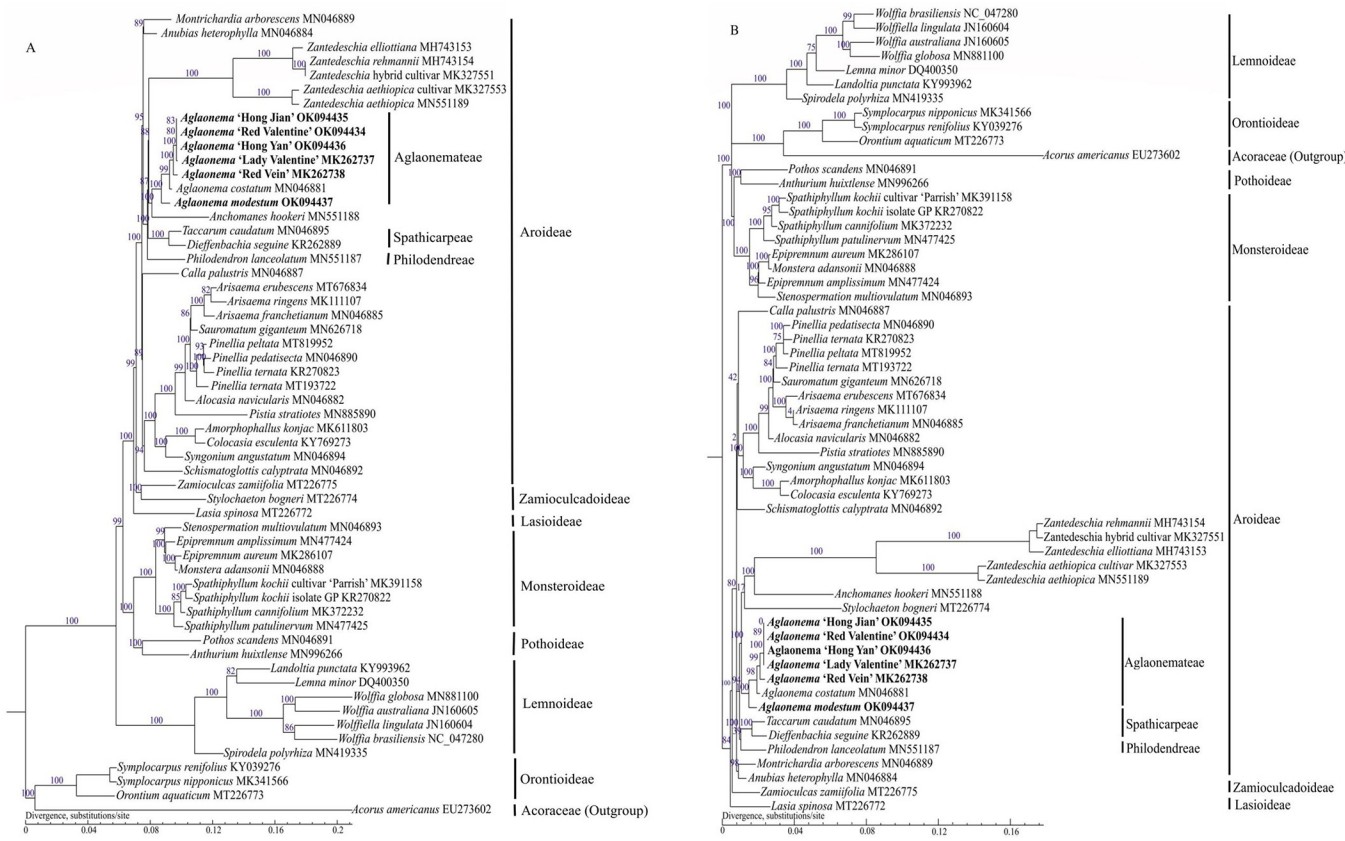

**Fig 10. Phylogenetic analyses of 56 Araceae chloroplast genomes with maximum likelihood, *Acorus americanus* was the outgroup taxa.** (A) The phylogenetic tree was constructed by the complete chloroplast genome sequences. (B) The phylogenetic tree was constructed using protein coding genes sequences. The six newly sequenced *Aglaonema* chloroplast genomes in this study were in bold.

Furthermore, remarkable variations among the six *Aglaonema* species and variegated culti-vars were gene loss and intron loss events. Both variegated cultivars, 'Lady Valentine' and 'Red Vein', had *ycf68* and one intron in *trnG-UCC*, respectively, while the other four chloroplast genomes of *A. modestum*, 'Hong Jian', 'Red Valentine', and 'Hong Yan', lost *ycf68* and introns in *trnG-UCC* (S3 Table and S1 Fig). Additionally, *trnS-CGA*, *trnS-GGA* and *trnT-GGU* were missing in the chloroplast genomes of 'Lady Valentine' and 'Red Vein', but these three tRNA genes showed one copy in the other four sequenced *Aglaonema* chloroplast genomes (Tables 2 and S3). In contrast to the present study, certain events of gene loss, intron loss, and gene duplication were also reported within Araceae chloroplast genomes; for example, gene loss events involved *rpl23*, *rpl2*, *trnL-CCA*, *trnG-GCC*, *accD* and *psbE* in the genus *Amorphophallus* [38].

Another variation among the tested *Aglaonema* species and variegated cultivars was the location of the boundaries among the four chloroplast genome regions. Location of the bound-aries, particularly boundaries of the IR contraction and expansion, has been considered an important evolutionary event leading to gene duplication or reduction of duplicate genes to a single copy [6,9,39]. In contrast to previous studies, duplication of *ycf1* and *rps15* has been reported in the subfamily Lemnoideae [39], and conversion of duplicate genes to single-copy genes has occurred in *rpl2* and *rpl23* in *Anchomanes hookeri* of the family Araceae [6]. Our results revealed the duplication of *rpl2* in IR/LSC boundary regions and integration of *trnH-GUG* into the IRa region (Fig 6), a finding that was also consistent with previous studies

findings [6,9]. Additionally, our findings showed that *trnN-GUU* was duplicated in IR/SSC borders in two chloroplast genomes of 'Lady Valentine' and 'Red Vein', respectively, and *ycf1* was duplicated in IR/SSC borders in the other five *Aglaonema* genomes (Fig 6). The same phenomenon exists in some species of the family Araceae [39–41].

Chloroplast genomes are rich in SSRs, long repeats, highly divergent regions, SNPs and indels [39–41]. They have been extensively used to identify species and cultivars [36] and for phylogenetic analyses [37,41]. In the present study, SSR detection results were the same as most other Araceae plants; for example, most SSRs were mononucleotide repeats and usually comprised short repeats of A/T [31,41,42]. Additionally, divergent analyses implemented by mVISTA and nucleotide diversity revealed 9 highly divergent regions among 7 *Aglaonema* chloroplast genomes, including *rnH-GUG-CDS1_psbA*, *trnS-GCU_trnS-CGA-CDS1*, *rps4-trnT-UGU*, *trnF-GAA-ndhJ*, *petD-CDS2-rpoA*, *ycf1-ndhF*, *rps15-ycf1-D2*, *ccsA-ndhD*, and *trnY-GUA-trnE-UUC* (Figs 7 and 8). In contrast to previous studies, six highly divergent regions (*trnH-GUG_psbA*, *rps4-trnT-UGU*, *trnF-GAA-ndhJ*, *rps15-ycf1*, *ccsA-ndhD*, and *petD-rpoA*) were used as DNA barcodes in Araceae plants or were in the marker development of DNA barcodes [9,41]. We suggest that the remaining three highly divergent regions of *trnS-GCU_trnS-CGA-CDS1*, *ycf1-ndhF*, and *trnY-GUA-trnE-UUC* can be used as special DNA barcodes for *Aglaonema* species and cultivar identification and phylogenetic analyses in the future.

Because 'Red Valentine' and 'Lady Valentine' were challenging to differentiate by their leaf morphological characteristics, developing molecular markers to identify these two *Aglaonema* cultivars is crucial. In a previous study, correlations among SNPs, indels, and repeats were investigated among 27 species from 7 subfamilies of Araceae, and strong/very strong correlations in most comparisons were observed [43]. Notably, in the present study, 14 SNPs and 1 insertion were found between these two cultivar genomes. These 14 SNPs were all located in *psaA* (S10 Table), and 1 insertion was found in *rps12* (S11 Table). Hence, *psaA* and *rps12* sequences can be used to differentiate these two cultivars. Additionally, the other two cultivar pairs, 'Red Valentine' vs. 'Red Vein' and 'Lady Valentine' vs. 'Red Vein', also contained many SNPs and indels (Tables 3, S10 and S11). These SNPs and indels could be used to identify these three variegated cultivars. However, both 'Red Valentine' vs. 'Hong Jian' and 'Red Valentine' vs. 'Hong Yan' had no SNPs/indels. These two comparisons indicated that the chloroplast genomes had not undergone recombination in two tissue culture-induced mutations of 'Red Valentine'. The two mutations, 'Hong Jian' and 'Hong Yan', may be induced by nuclear gene variation and complex biological processes. The molecular regulation mechanisms of leaf shape and leaf color variations of 'Hong Jian' and 'Hong Yan' require further investigation.

In the present study, the Ka/Ks ratio showed low rates of evolution (Ka/Ks < 1) for most protein coding genes (74 of 79 genes) among the 16 Araceae chloroplast genomes tested. These findings agreed with those of other high plant chloroplast genomes [9,33,43–45]. However, our results contradicted a study in which majority (62 of 71 genes) had a Ka/Ks ratio of > 1, suggesting that these genes were positively selected within 14 Araceae species [35]. We demonstrated that *rpl2*, *rps2*, *rps3*, *ycf1* and *ycf2* were under positive selection among the 16 Araceae chloroplast genomes tested (S9 Table). By comparison, previous studies have revealed that *rps3*, *ycf1* and *ycf2* with positive selection in high plants may be very common [9,33,43,44]; *ycf1* and *ycf2* have been identified under positive selection in *Eruca sativa* [44]; *ycf1* and *ycf2* have also been reported under positive selection in *Pothos scandens* and Monsteroideae species of Araceae [9,33]; and *rps3*, *ycf1* and *ycf2* have been identified under positive selection in four *Zingiber* species [45].

Many studies have used complete chloroplast genomes or protein coding genes for phylogenetic studies in the family Araceae [6,9,31,33–35,40,41,46]. Specifically, chloroplast *trnL-trnF*

has been used in Lemnaceae and Araceae phylogenetic relationship studies [47]. Additionally, previous molecular research based on complete chloroplast genomes identified eight subfamilies (excluding Gymnostachydoideae) within the family Araceae: Aroideae, Lasioideae, Lemnoideae, Monsteroideae, Orontioideae, Pothoideae, and Zamioculcadoideae [46]. Genetic relationships among the subfamilies of Orontioideae, Lemnoideae, Pothoideae and Monsteroideae were strongly supported [46]. Our phylogenetic tree obtained by whole chloroplast genomes was in broad agreement with previous studies [6,9,31,33,46]. However, the two phylogenetic trees, constructed using whole chloroplast genomes and protein coding genes, exhibited few inconsistencies in this study—for example, the shifting position of *Stylochaeton* in subfamily Aroideae in the phylogenetic tree based on protein coding genes (Fig 10A and 10B). The bootstrap value (17%) was too low in the latter tree to draw any meaningful conclusions about genetic relationships in subfamily Aroideae (Fig 10B). The reason for the shifting position of *Stylochaeton* may be (1) that the protein coding gene sequences were insufficient to resolve the relationships between *Stylochaeton* and the rest of subfamily Aroideae with strong support and (2) that more *Stylochaeton* and other Aroideae species chloroplast genomes must be sequenced to solve the uncertainties. Additionally, the position of *Acorus* contrasted that in previous studies [6,9,33]. Both of our trees indicated that *Acorus* was sister to the genera *Orontium* + *Symplocarpus* with strong support (bootstrap values = 100%) (Fig 10). Therefore, *Acorus* requires more chloroplast data or nuclear genomes to resolve its position.

## Conclusions

In this study, the complete chloroplast genomes of *A. modestum* and five variegated cultivars of *Aglaonema* were sequenced and assembled. The six chloroplast genomes were highly conserved in terms of the gene content, codon usage, and genome structure but also distinguished the difference between IR-SSC boundary regions. 'Red Vein' differed significantly from the others in long repeat number and type. Comparative analyses of the chloroplast genomes identified 9 divergent hotspots (Pi > 0.021) with potential application as DNA barcodes. The Ka/Ks ratio analyses among 16 Araceae chloroplast genomes showed that *rpl2*, *rps2*, *rps3*, *ycf1* and *ycf2* were under positive selection. Phylogenetic trees based on whole chloroplast genomes and protein-coding genes strongly supported monophyletic *Aglaonema*. These assembled chloroplast genomes provided potential genomic resources and helped the identification and utilization of *Aglaonema* germplasm resources.

## Supporting information

**S1 Fig. Chloroplast genome maps of the five variegated cultivars of Aglaonema in this study.** Genes shown inside the circle are transcribed clockwise, and those outside are transcribed counterclockwise. The gray arrowheads indicate the direction of the genes. Different genes are color coded. The innermost darker gray corresponds to the GC content, whereas the lighter gray corresponds to the AT content. The inner circle also indicates that the chloroplast genome contains a large single copy (LSC) region, a small single copy (SSC) region and two copies of the inverted repeat (IRA and IRB). (A) 'Red Valentine', (B) 'Hong Yan', (C) 'Hong Jian', (D) 'Lady Valentine', and (E) 'Red Vein'. * in (D) and (E) indicates *ycf68* and *trnI-CAU* only present in the IR regions of the chloroplast genomes of 'Lady Valentine' and 'Red Vein', respectively.
(DOC)

**S1 Table. Chloroplast genomes used for Ka/Ks and phylogenetic analyses.**
(XLSX)

S2 Table. Features of the six newly sequenced chloroplast genomes of the genus Aglaonema.
(DOCX)

S3 Table. Annotations of the six newly sequenced chloroplast genomes in the genus Aglaonema.
(DOCX)

S4 Table. Genes with introns in the six newly sequenced chloroplast genomes of the genus Aglaonema.
(XLS)

S5 Table. Codon usage of the protein-coding genes in the six newly sequenced chloroplast genomes of the genus Aglaonema.
(XLS)

S6 Table. Long repeat distribution among the six newly sequenced chloroplast genomes of the genus Aglaonema.
(XLSX)

S7 Table. SSR distribution among the six newly sequenced chloroplast genomes of the genus Aglaonema.
(XLSX)

S8 Table. Nucleotide diversity values among seven chloroplast genomes of the genus Aglaonema.
(XLSX)

S9 Table. Ka/Ks ratios among 16 chloroplast genomes of Araceae.
(XLS)

S10 Table. SNPs detected in protein coding and intergenic regions among seven Aglaonema chloroplast genomes.
(XLS)

S11 Table. Indels detected among seven Aglaonema chloroplast genomes.
(XLS)

## Author Contributions

**Conceptualization:** Dong-Mei Li.

**Data curation:** Dong-Mei Li.

**Formal analysis:** Dong-Mei Li, Bo Yu, Dan Huang.

**Funding acquisition:** Dong-Mei Li.

**Project administration:** Dong-Mei Li.

**Resources:** Dong-Mei Li.

**Software:** Dong-Mei Li.

**Supervision:** Dong-Mei Li, Gen-Fa Zhu.

**Validation:** Dong-Mei Li, Gen-Fa Zhu.

**Visualization:** Dong-Mei Li, Gen-Fa Zhu.

**Writing – original draft:** Dong-Mei Li.

**Writing – review & editing:** Dong-Mei Li.

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
