## [Decision Letter · Decision Letter 0]

25 Jul 2022

PONE-D-22-15194Comparative chloroplast genomes and phylogenetic relationships of Aglaonema modestum and five foliar variegation cultivars of Aglaonema (Araceae)

PLOS ONE

Dear Dr. Li,

Thank you for submitting your manuscript to PLOS ONE. After careful consideration, we feel that it has merit but does not fully meet PLOS ONE’s publication criteria as it currently stands. Therefore, we invite you to submit a revised version of the manuscript that addresses the points raised during the review process.

The paper contains valuable results for the Aglaonema community but should be carefully edited and corrected to improve the language and eliminate grammatic errors and style issues.

We look forward to receiving your revised manuscript.

Kind regards,

Chunxian Chen, Ph.D.

Academic Editor

PLOS ONE

Journal Requirements:

    "No." 

Additional Editor Comments:

In this study, chloroplast genomes were generated and used to elucidate the phylogenetic relationships of cultivated Aglaonema in South China. The genomes are valuable resources for Aglaonema. Substantial improvement and edition in language may be needed, with attention to many long sentences and redundant content. Here are a few examples and the authors should go through the entire paper.

Title:

Replace “foliar variegation” (a noun) with “variegated” (an adjective) before “cultivar” (a noun). They refer to the same botanical character. Remove the family name at the end, which is unnecessary.

Likewise, replace all “foliar variegation” with “variegated” in the entire paper if the two words are used as an adjective before “cultivar”.

Abstract:

Line 14-16: Reorganize the sentence and remove the redundant word Aglaonema. All words in cultivar names should be capitalized as they are specific nouns.

Line 20-23, this long sentence contains many grammatical errors. For example, “that” should be ”these”. “were significant different” should be “were significantly different” or “had significant differences”. Should it be “distinctive” (typical) or “distinct” (different)?

Line 22, remove “A. “ before ‘Red vein’ and all cultivar names, unnecessary and awkward.

Line 28, How are they “positive selection”? are they just spontaneous mutations accumulated in the genomes?

Introduction:

Line 45: “has started since 1930s and soon” should change to “started in 1930s and”

Line 47: remove the hyphen in “sports-selected”

Line 50-51, “Aglaonema species and cultivars identifications” may be better as “identification of Aglaonema species and cultivars”

……

M&M

Line 82-84, move this section elsewhere following the journal’s instruction.

Line 85, should be “materials”

Line 86-87, remove all “A.” before all cultivars and capitalize all cultivar names

……

Reviewers' comments:

Reviewer's Responses to Questions

**Comments to the Author**

1. Is the manuscript technically sound, and do the data support the conclusions?

Reviewer #1: Partly

Reviewer #2: Partly

2. Has the statistical analysis been performed appropriately and rigorously? 

Reviewer #1: N/A

Reviewer #2: Yes

3. Have the authors made all data underlying the findings in their manuscript fully available?

Reviewer #1: Yes

Reviewer #2: Yes

4. Is the manuscript presented in an intelligible fashion and written in standard English?

Reviewer #1: Yes

Reviewer #2: No

5. Review Comments to the Author

Reviewer #1: The authors reported chloroplast genomes of Aglaonema modestum and five Aglaonema foliar variegation cultivars. The results enriched chloroplast genome data of the genus, thus is of some value. But I don't think it is suitable for publication is its present form, before the authors make a substantial revision.

1. Separate Discussion section from the Result section, and strengthen it to elucidate the findings.

2. Introduce clearly the relationship between Aglaonema modestum and five Aglaonema foliar variegation cultivars selected.

3. I can not agree with the conclusion "Phylogenetic analysis indicated that A. modestum was close to A. costatum, and that the five foliar variegation cultivars could be clearly classified." To draw such a conclusion, more samples of each species/cultivar are needed.

Reviewer #2: In this manuscript, authors compared chloroplast genome on one green species and five foliar variegation cultivars of Aglaonema. Authors have done detail comparison and characterization of the six genomes. The study might be interesting for publication in PLOS ONE. However, the manuscript does have several issues which I think need to be addressed.

Major

1. Authors point out that it is difficulty to identify Aglaonema species and cultivars based on leaf morphology. It seems there is no clear answer to that question. Phylogenetic analysis just shows the genetic relationship among Aglaonema species and cultivars. It could be very useful if authors provide molecular markers to identify or separate Aglaonema species and cultivars, especially for ‘two mutations Hong yan’ and ‘Hong jian’, as well as ‘Red valentine.

2. There is limited discussion, and most are presented results in ‘Results and Discussion’. For example, in line 157, “Interestingly, both A. ‘Red valentine’ and A. ‘Hong jian’ were with same genome size, which was 158 bp smaller than that of A. ‘Hong yan’. However, there is no explanation or discussion why authors think this is an interesting result. I suggested to separate ‘Results and Discussion’ into two sections: Result’, ‘Discussion’.

Minor

3. line 52-56, the background information of the plant materials (Fig.1) should be moved to the section ‘Plant Material’.

4. Fig2, except genome sizes, there is no other information, such as genes, GC content for six sequenced chloroplast genomes of genus Aglaonema.

5. Fig3 B, it should be interesting to show a cluster for the six sequenced chloroplast genomes. Heatmap shows they are all the same and no difference.

6. PLOS authors have the option to publish the peer review history of their article (what does this mean?). If published, this will include your full peer review and any attached files.

Reviewer #1: No

Reviewer #2: No

---

## [Author Response · Author response to Decision Letter 0]

19 Aug 2022

Journal Requirements:

Reponse: We revised the style requirements one by one.

Reponse: We checked the ‘Funding Information’ section and found the correct grant numbers for the awards. We added “Guangdong Province Modern Agriculture Industry Technical System-Flower Innovation Team Construction Project (2021KJ121)”, because if the manuscript is accepted, the publication fee is from this project (2021KJ121). 

 When we returned the manuscript, we found no ‘Financial Disclosure’ section in the questions. Hence, we wrote ‘Funding’ information behind ‘Data Availability Statement’.

Funding: This research was financially supported by the Guangdong Basic and Applied Basic Foundation Project (No. 2021A1515010893), Special Financial Fund of Foshan-High-level Guangdong Agricultural Science and Technology Demonstration City Project (2022), and Guangdong Province Modern Agriculture Industry Technical System-Flower Innovation Team Construction Project (2021KJ121). The funders had no role in study design, data collection and analysis, decision to publish, or preparation of the manuscript.

    "No." 

Reponse: The authors have declared that no competing interests exist.

Reponse: The six newly assembled chloroplast genomes in this study have been deposited in GenBank with accession numbers MK262737, MK262738, OK094434, OK094435, OK094436, and OK094427.

Additional Editor Comments:

In this study, chloroplast genomes were generated and used to elucidate the phylogenetic relationships of cultivated Aglaonema in South China. The genomes are valuable resources for Aglaonema. Substantial improvement and edition in language may be needed, with attention to many long sentences and redundant content. Here are a few examples and the authors should go through the entire paper.

Title:

Replace “foliar variegation” (a noun) with “variegated” (an adjective) before “cultivar” (a noun). They refer to the same botanical character. Remove the family name at the end, which is unnecessary.

Likewise, replace all “foliar variegation” with “variegated” in the entire paper if the two words are used as an adjective before “cultivar”.

Reponse: Yes, we agreed and revised. 

Abstract:

Line 14-16: Reorganize the sentence and remove the redundant word Aglaonema. All words in cultivar names should be capitalized as they are specific nouns.

Reponse: Yes, we agreed and revised. 

Line 20-23, this long sentence contains many grammatical errors. For example, “that” should be ”these”. “were significant different” should be “were significantly different” or “had significant differences”. Should it be “distinctive” (typical) or “distinct” (different)?

Reponse: line 20, “that” should not be ”these”. We revised as following: but the IR-SSC boundary regions were significantly different, and ‘Red Vein’ had a distinct long repeat number and type frequency. 

We agreed with the revisions of line 21-23. “were significant different” should be “were significantly different” . “distinctive” should be “distinct”.

Line 22, remove “A. “ before ‘Red vein’ and all cultivar names, unnecessary and awkward.

Reponse: Yes, we agreed and revised. 

Line 28, How are they “positive selection”? are they just spontaneous mutations accumulated in the genomes?

Reponse: Positive selection was found for rpl2, rps2, rps3, ycf1 and ycf2 based on the analyses of Ka/Ks ratios among 16 Araceae chloroplast genomes.

Introduction:

Line 45: “has started since 1930s and soon” should change to “started in 1930s and”

Reponse: Yes, we agreed and revised. 

Line 47: remove the hyphen in “sports-selected”

Reponse: Yes, we agreed and revised. 

Line 50-51, “Aglaonema species and cultivars identifications” may be better as “identification of Aglaonema species and cultivars”

……

Reponse: Yes, we agreed and revised. 

M&M

Line 82-84, move this section elsewhere following the journal’s instruction.

Reponse: Yes, we agreed and revised. 

Line 85, should be “materials”

Reponse: Yes, we agreed and revised. 

Line 86-87, remove all “A.” before all cultivars and capitalize all cultivar names

……

Reponse: Yes, we agreed and revised. We improved the English writing quality of the manuscript with help from American Journal Experts (AJE). 

Reviewers' comments:

Reviewer's Responses to Questions

Comments to the Author

1. Is the manuscript technically sound, and do the data support the conclusions?

Reviewer #1: Partly

Reviewer #2: Partly

2. Has the statistical analysis been performed appropriately and rigorously?

Reviewer #1: N/A

Reviewer #2: Yes

3. Have the authors made all data underlying the findings in their manuscript fully available?

Reviewer #1: Yes

Reviewer #2: Yes

4. Is the manuscript presented in an intelligible fashion and written in standard English?

Reviewer #1: Yes

Reviewer #2: No

Reponse: We have improved the English writing quality of the manuscript with help from American Journal Experts (AJE). 

5. Review Comments to the Author

Reviewer #1: The authors reported chloroplast genomes of Aglaonema modestum and five Aglaonema foliar variegation cultivars. The results enriched chloroplast genome data of the genus, thus is of some value. But I don't think it is suitable for publication is its present form, before the authors make a substantial revision.

1. Separate Discussion section from the Result section, and strengthen it to elucidate the findings.

Reponse:Yes, we agreed. We separated ‘Results and Discussion’ into two sections: ‘Result’ and ‘Discusssion’. Details of ‘Discusssion’ can be found in the revised manuscript.

2. Introduce clearly the relationship between Aglaonema modestum and five Aglaonema foliar variegation cultivars selected.

Reponse: Yes, we introduced clearly the leaf morphological charateristics of A. modestum and five variegated cultivars in the section ‘Plant Material’ as following:

 A. modestum is erect with a dark green stem and leaves, which is not variegated (Fig. 1A). Both ‘Red Valentine’ and ‘Lady Valentine’ are variegated, with green and red along the midrib, along veins and throughout the leaf blade (Fig. 1B-1C). The leaves of ‘Hong Jian’ are narrow, oblong, dark pink to dark red, accounting for approximately 75% to 90% of the leaf area, and the shape of the apex of the leaves is strongly acute, with green and dark yellow green blotches at the margin and throughout the leaf blade (Fig. 1D). The leaves of ‘Hong Yan’ are ovate-cordate, dark red, accounting for approximately 95% of the leaf area, and the shape of the apex of the leaves is obtuse to moderately acute, with green blotches at the leaf margin (Fig. 1E). The leaves of ‘Red Vein’ are dark green with pink to red along the midrib and veins (Fig. 1F). 

3. I can not agree with the conclusion "Phylogenetic analysis indicated that A. modestum was close to A. costatum, and that the five foliar variegation cultivars could be clearly classified." To draw such a conclusion, more samples of each species/cultivar are needed.

Reponse: Thank you very much for this suggestion. We revised the "Phylogenetic analyses” section and revised the conclusion as following:

In this study, two phylogenetic trees were constructed by ML method using whole chloroplast genomes and protein coding genes, respectively (Fig 10A-10B). A. americanus was classified in family Acoraceae [32], which was used as the outgroup. The 7 subfamilies within Araceae, including Aroideae, Lasioideae, Lemnoideae, Monsteroideae, Orontioideae, Pothoideae, and Zamioculcadoideae, could be stongly identified by the whole chloroplast genomes with bootstrap values 89-100% (Fig 10A). In contrast, the phylogenetic tree constructed by protein coding genes showed moderate (bootstrap value = 84%) to strong identifications (bootstrap value = 100%) (Fig 10B). Within subfamily Aroideae, Aglaonema was strongly supported as monophyletic (bootstrap values = 100%) (Fig 10A-10B). Aglaonema was seen as either sister to Anchomanes (bootstrap value = 100%), and Aglaonema + Anchomanes and Zantedeschia were strongly supported as sister genera (bootstrap value = 88%) (Fig 10A), or formed a poorly-supported clade with Stylochaeton (bootstrap value = 17%) that was sister to genera Anchomanes + Zantedeschia (bootstrap value = 100%) (Fig 10B). Within genus Aglaonema, ‘Hong Jian’, ‘Red Valentine’, ‘Hong Yan’, and ‘Lady Valentine’ were clustered together with moderate to strong support (bootstrap values = 80-100%), and ‘Red Vein’ was sister to these four variegated cultivars with strong support (bootstrap value = 100%) (Fig 10A). A. costatum was sister to these five variegated cultivars with strong support (bootstrap value = 99%). A. modestum was sister to the clade of “A. costatum + five variegated cultivars” with strong support (bootstrap value = 100%) based on complete chloroplat genomes (Fig 10A). These seven Aglaonema species and cultivars showed similar topologies based on protein coding genes, but ‘Hong Jian’ and ‘Red Valentine’ were poorly supported in the ML analysis (bootstrap value = 0) (Fig 10B). The other five Aglaonema species and cultivars were strongly supported in ML analysis based on protein coding genes (bootstrap values = 89-100%) (Fig 10B). 

Reviewer #2: In this manuscript, authors compared chloroplast genome on one green species and five foliar variegation cultivars of Aglaonema. Authors have done detail comparison and characterization of the six genomes. The study might be interesting for publication in PLOS ONE. However, the manuscript does have several issues which I think need to be addressed.

Major

1. Authors point out that it is difficulty to identify Aglaonema species and cultivars based on leaf morphology. It seems there is no clear answer to that question. Phylogenetic analysis just shows the genetic relationship among Aglaonema species and cultivars. It could be very useful if authors provide molecular markers to identify or separate Aglaonema species and cultivars, especially for ‘two mutations Hong yan’ and ‘Hong jian’, as well as ‘Red valentine.

Reponse: Yes, we provided molecular markers to identify or separate among A. modestum, A. costatum, ‘Red valentine’, ‘Lady Valentine’, and ‘Red Vein’ (Table 3, S10 Table, and S11 Table).

It is difficult to identify two cultivars ‘Red valentine’ and ‘Lady Valentine’ by their leaf morphological charateristics. In this study, it was worth noting that 14 SNPs and 1 insertion were found between these two cultivars genomes. These 14 SNPs were all located in psaA (S10 Table), and 1 insertion was found in rps12 (S11 Table). Hence, psaA and rps12 sequences can be used to differentiate these two cultivars. However, two mutations ‘Hong yan’ and ‘Hong jian’, as well as ‘Red valentine, had no SNPs/indels, which indicated that the chloroplast genomes had not undergone recombination in two tissue-culture induced mutations of ‘Red Valentine’. The two mutations, ‘Hong Jian’ and ‘Hong Yan’ may be induced by nuclear gene variation and complex biological processes. The molecular regulation mechanisms of leaf shape and leaf colour variations of ‘Hong Jian’ and ‘Hong Yan’, need further investigation.

2. There is limited discussion, and most are presented results in ‘Results and Discussion’. For example, in line 157, “Interestingly, both A. ‘Red valentine’ and A. ‘Hong jian’ were with same genome size, which was 158 bp smaller than that of A. ‘Hong yan’. However, there is no explanation or discussion why authors think this is an interesting result. I suggested to separate ‘Results and Discussion’ into two sections: Result’, ‘Discussion’.

Reponse: Yes, we agreed. We separated ‘Results and Discussion’ into two sections: ‘Result’ and ‘Discusssion’. Details of ‘Discusssion’ can be found in the revised manuscript.

Minor

3. line 52-56, the background information of the plant materials (Fig.1) should be moved to the section ‘Plant Material’.

Reponse: Yes, we agreed. We moved Fig.1 to the section ‘Plant Material’. 

4. Fig2, except genome sizes, there is no other information, such as genes, GC content for six sequenced chloroplast genomes of genus Aglaonema.

Reponse: We revised Fig.2 to display chloroplast genome map of A. modestum and added the other five sequenced chloroplast genomes maps in S1 Fig. The genes, GC content and other genome information were showed in Table 1, S1 Table, and S2 Table.

5. Fig3 B, it should be interesting to show a cluster for the six sequenced chloroplast genomes. Heatmap shows they are all the same and no difference.

Reponse: Yes, we agreed. As we draw the Fig 3B using RSCU values with two decimal data, it looked no difference. We revised Fig 3B using all four decimal data as following, which existed differences. 

6. PLOS authors have the option to publish the peer review history of their article (what does this mean?). If published, this will include your full peer review and any attached files.

Do you want your identity to be public for this peer review? For information about this choice, including consent withdrawal, please see our Privacy Policy.

Reviewer #1: No

Reviewer #2: No

Reponse: We revised the figures by PACE on line.

---

## [Editor Report · Decision Letter 1]

22 Aug 2022

Comparative chloroplast genomes and phylogenetic relationships of Aglaonema modestum and five variegated cultivars of Aglaonema

PONE-D-22-15194R1

Dear Dr. Li,

We’re pleased to inform you that your manuscript has been judged scientifically suitable for publication and will be formally accepted for publication once it meets all outstanding technical requirements.

Kind regards,

Chunxian Chen, Ph.D.

Academic Editor

PLOS ONE
---

## [Editor Report · Acceptance letter]

25 Aug 2022

PONE-D-22-15194R1 

Comparative chloroplast genomes and phylogenetic relationships of *Aglaonema modestum* and five variegated cultivars of *Aglaonema*

Dear Dr. Li:

I'm pleased to inform you that your manuscript has been deemed suitable for publication in PLOS ONE. Congratulations! Your manuscript is now with our production department. 

Kind regards, 

on behalf of

Dr. Chunxian Chen 

Academic Editor

PLOS ONE